

# Ocean Color Algorithm for the Retrieval of the Particle Size Distribution and Carbon-Based Phytoplankton Size Classes Using a Two-Component Coated-Spheres Backscattering Model

Tihomir S. Kostadinov[1], Lisl Robertson Lain[2], Christina Eunjin Kong[3], Xiaodong Zhang[4], Stéphane Maritorena[5], Stewart Bernard[6], Hubert Loisel[7], Daniel S.F. Jorge[8], Ekaterina Kochetkova[9], Shovonlal Roy[10], Bror Jonsson[11], Victor Martinez-Vicente[12], and Shubha Sathyendranath[13]

[1]Department of Liberal Studies, California State University San Marcos, 333 S. Twin Oaks Valley Rd., San Marcos, CA 92096, USA

[2]Earth Observation, Smart Places, CSIR; formerly with University of Cape Town, Department of Oceanography, Cape Town, South Africa

[4]Division of Marine Science, School of Ocean Science and Engineering, The University of Southern Mississippi, Stennis Space Center, MS 39529, USA

[5]Earth Research Institute, University of California at Santa Barbara, Santa Barbara, CA 93106-3060, USA

[6]SANSA, Enterprise Building, Mark Shuttleworth Street, Innovation Hub, Pretoria 0087, South Africa

[7,8]Laboratoire d'Océanologie et de Géosciences, Université du Littoral-Côte-d'Opale, Université Lille, CNRS, IRD, UMR 8187, LOG, 32 avenue Foch, Wimereux, France

[9]Department of Earth & Environmental Science, Hayden Hall, University of Pennsylvania, 240 South 33rd St., Philadelphia, PA 19104, USA

[10]Department of Geography and Environmental Science, University of Reading, Reading, United Kingdom

[3,11,12,13]National Centre for Earth Observation, Plymouth Marine Laboratory, UK

**Correspondence:** Tihomir S. Kostadinov (tkostadinov@csusm.edu)

**Abstract.** The particle size distribution (PSD) of suspended particles in near-surface seawater is a key property linking biogeochemical and ecosystem characteristics with optical properties that affect ocean color remote sensing. Phytoplankton size affects their physiological characteristics and ecosystem and biogeochemical roles, e.g. in the biological carbon pump, which has an important role in the global carbon cycle and thus climate. It is thus important to develop capabilities for measure-
ment and predictive understanding of the structure and function of oceanic ecosystems, including the PSD, phytoplankton size classes (PSCs) and phytoplankton functional types (PFTs). Here, we present an ocean color satellite algorithm for the retrieval of the parameters of an assumed power-law PSD. The forward optical model considers two distinct particle populations (particle assemblage categories) — phytoplankton and non-algal particles (NAP). Phytoplankton are modeled as coated spheres following the Equivalent Algal Populations (EAP) framework, and NAP are modeled as homogeneous spheres. The forward
model uses Mie and Aden-Kerker scattering computations, for homogeneous and coated spheres (for phytoplankton and NAP, respectively) to model the total particulate spectral backscattering coefficient as the sum of phytoplankton and NAP backscattering. The PSD retrieval is achieved via Spectral Angle Mapping (SAM) which uses backscattering end-members created by the forward model. The PSD is used to retrieve size-partitioned absolute and fractional phytoplankton carbon concentrations (i.e. carbon-based PSCs), as well as particulate organic carbon (POC), using allometric coefficients. The EAP-based formu-
lation allows for the estimation of chlorophyll-a concentration via the retrieved PSD, as well as the estimation of the percent




of backscattering due to NAP vs. phytoplankton. The PSD algorithm is operationally applied to the merged Ocean Colour Climate Change Initiative (OC-CCI) v5.0 ocean color data set. Results of an initial validation effort are also presented, using PSD, POC, and pico-phytoplankton carbon *in-situ* measurements. Validation results indicate the need for an empirical tuning for the absolute phytoplankton carbon concentrations; however these results and comparison with other phytoplankton car-

bon algorithms are ambiguous as to the need for the tuning. The latter finding illustrates the continued need for high-quality, consistent, large global data sets of phytoplankton carbon and related variables to facilitate future algorithm improvements.

## 1 Introduction

Oxygenic photosynthesis by marine phytoplankton is a critical planetary scale process supplying solar energy to the biosphere by fixing inorganic carbon; it is responsible for roughly half of global annual net primary productivity (e.g. Field et al. (1998)).

Ocean ecosystems play a key role in Earth's carbon cycle and climate by affecting atmospheric $CO_2$ via the biological carbon pump, which sequesters some of the fixed carbon to the deeper ocean for longer time scales (e.g. Eppley and Peterson (1979); Chisholm (2000); Henson et al. (2011); Boyd et al. (2019); Brewin et al. (2021)). The biological pump is influenced by the structure and function of oceanic ecosystems (e.g. Falkowski et al. (1998); Siegel et al. (2014)); therefore, mechanistic, predictive understanding of ocean ecosystems is of high priority to Earth systems and climate research (e.g. Buesseler and

Boyd (2009); Siegel et al. (2016)). Satellite remote sensing of ocean color is a key tool for the global characterization of ocean ecology (e.g. Siegel et al. (2013)). This has led to large efforts to elucidate biological pump mechanisms using multiple platforms, including satellites, e.g. the EXPORTS Program (Siegel et al., 2016).

Phytoplankton cell size (diameters varying from $\approx 0.5\,\mu m$ to $> 50\,\mu m$ (e.g. Clavano et al. (2007)) is an important trait that affects multiple phytoplankton characteristics (Marañón (2015)), as well as sinking rates (e.g. Falkowski et al. (1998); Burd

and Jackson (2009); Stemmann and Boss (2012); Siegel et al. (2014)). Phytoplankton size classes (PSCs) thus tend to closely correspond to phytoplankton functional types (PFTs, e.g., Quéré et al. (2005)). Importantly, phytoplankton cells also affect the inherent optical properties (IOPs) (e.g. absorption and backscattering coefficients) of the water column in a size-dependent manner (e.g. Mobley et al. (2002); Morel and Bricaud (1986); Stramski and Kiefer (1991); Kostadinov et al. (2009)). This is because particle size (relative to the incident light wavelength) is one of the governing variables affecting the magnitude and

spectral shape of light scattering and absorption caused by a particle (e.g. Bohren and Huffman (1983)). Therefore the particle size distribution (PSD) of phytoplankton (and other suspended particles in seawater) is a key property affecting both optical properties and cellular physiological and biogeochemical properties, i.e. it is a fundamental property linking ocean color remote sensing and ecosystem/biogeochemical characteristics. The size distribution of particles suspended in near surface ocean waters is often described as a power law, given in differential form as follows (e.g. Bader (1970); Sheldon et al. (1972); Jonasz (1983);

Boss et al. (2001); Twardowski et al. (2001); Kostadinov et al. (2009); Roy et al. (2017)):

$$\frac{dN_T}{dD} = N(D) = N_0 \left( \frac{D}{D_0} \right)^{-\xi} \tag{1}$$



where $N$ [m$^{-4}$] is the differential number concentration of particles per unit volume seawater and per bin width of particle diameter, $N_0 = N(2 \, \mu\mathrm{m})$ is the particle number concentration at a reference diameter, here $D_0 = 2 \, \mu\mathrm{m}$, $D$ is particle diameter, and $\xi$ is the power-law slope of the PSD. Equation 1 has to be integrated over a given diameter range to get the total volumetric
particle concentration in that range, $N_T$, [m$^{-3}$].

Ocean color is quantified by the spectral shape and magnitude of the remote-sensing reflectance, $R_{rs}(\lambda)$ [sr$^{-1}$], where $\lambda$ is the wavelength of light *in vacuo*. The Kostadinov-Siegel-Maritorena 2009 (KSM09, Kostadinov et al. (2009)) algorithm retrieves the parameters of an assumed power-law PSD ($\xi$ and $N_0$ in Eq. 1) from ocean color remote-sensing observations, using the spectral shape (Loisel et al., 2006) and magnitude of the particulate backscattering coefficient, $b_{bp}(\lambda)$ [m$^{-1}$]. $b_{bp}(\lambda)$ can be
retrieved using existing inherent optical property (IOP) inversion algorithms; KSM09 uses the Loisel and Stramski (2000) IOP inversion. Subsequently, the retrieved PSD parameters allow the quantification of absolute and fractional PSCs — picoplankton, nanoplankton and microplankton, based on bio-volume (Kostadinov et al., 2010) or phytoplankton carbon (Kostadinov et al., 2016a) (henceforth TK16) via allometric relationships (Menden-Deuer and Lessard, 2000). Phytoplankton carbon (phyto C) is the key variable of interest for carbon cycle and climate studies and modeling, and TK16 (data set available — Kostadinov
et al. (2016b)) represents a relatively unique carbon-based approach among PSC/PFT algorithms (Mouw et al., 2017) as it is based on knowledge of the PSD and allometric relationships to get at size-partitioned phyto C. Roy et al. (2013, 2017) retrieve phytoplankton-specific PSD and size-partitioned phyto C, based on the phytoplankton absorption coefficient.

The KSM09 PSD algorithm (and the TK16 phyto C/PSC derived from it) is built on the assumption of a single population of particles (approximated by homogeneous spheres), representing backscattering due to the entire oceanic particle assemblage
— phytoplankton cells and non-algal particles (NAP). However, particle internal composition and shape influence its optical properties (e.g Quirantes and Bernard (2004, 2006)). Recent results suggest that the structural complexity of oceanic particles enhances backscattering significantly and can explain the so-called "missing backscattering" in the ocean (Organelli et al., 2018), i.e. the lack of optical closure between theoretically modeled and measured $b_{bp}$. Coated spheres (i.e. spheres consisting of concentric layers/shells of different material properties) can be used to better represent phytoplankton cells and their internal
heterogeneity and composition (e.g. Bernard et al. (2009); Robertson Lain and Bernard (2018)), and they have significantly enhanced backscattering compared to their homogeneous equivalents (Duforêt-Gaurier et al., 2018; Organelli et al., 2018).

Here, we introduce a major improvement of the KSM09 PSD algorithm. Two separate particle populations are modeled, living phytoplankton cells and NAP. Phytoplankton cells are modeled as coated spheres, following the Equivalent Algal Populations (EAP) framework (Bernard et al., 2009; Robertson Lain et al., 2014; Robertson Lain and Bernard, 2018). EAP explicitly
models intracellular chlorophyll concentration, $Chl_i$, as governing the imaginary index of refraction, and thus allows for bulk chlorophyll concentration (Chl) to be computed from a specific PSD. The coated sphere EAP-based model is useful to better represent specifically phytoplankton cells; however, not all backscattering particles are phytoplankton (Stramski et al., 2004), and in fact, sub-micron NAP even smaller than the smallest autotroph ($\approx 0.5 \, \mu\mathrm{m}$ in diameter) are critical for determining the spectral shape of $b_{bp}$, which is key for PSD retrieval with KSM09 and the algorithm presented here. Particles other than and
smaller than phytoplankton are likely to significantly contribute to backscattering (Stramski et al., 2004; Zhang et al., 2020), in spite of evidence that phytoplankton/larger particles contribute more than Mie theory predicts, based on homogeneous spheres





(e.g. Dall'Olmo et al. (2009)). Thus, a 2-component particle model is used here, separately modeling NAP as homogeneous spheres of wider size range than phytoplankton, so that bulk $b_{bp}$ of oceanic waters can be modeled (e.g. Stramski et al. (2001); Moutier et al. (2016); Duforêt-Gaurier et al. (2018)). NAP are modeled as having generally organic detrital composition, but

with some allowance for higher indices of refraction to account for minerogenic particle contributions. The PSD forward model can thus also produce a first-order estimate of POC, and the percent contribution of phytoplankton and NAP to $b_{bp}$.

Subsequent sections present details of the 2-component, EAP-based forward IOP model, the inversion methodology developed for operational application of the PSD algorithm, and the subsequent use of the satellite-derived PSD to retrieve derived products (following the methods of TK16 with some modifications), namely - absolute and fractional size-partitioned phy-

toplankton carbon (henceforth phyto C) (i.e. carbon-based PSCs), as well as Chl and POC estimates. The novel algorithm is applied operationally to monthly data from the multi-sensor merged OC-CCI v5.0 data set (Sathyendranath et al., 2019, 2021) — examples are shown in the manuscript, and the entire data set is publicly available and linked below (See Sec. 4). We then present and discuss an initial effort of validation of the new PSD algorithm and derived products using global compilations of PSD, pico-phytoplankton carbon and POC in-situ data. A comparison with other existing methods to retrieve phyto C is

presented. We also discuss algorithm uncertainties, assumptions and limitations as well as future work directions.

## 2 Data and Methods

### 2.1 Particle optical model input specification for Phytoplankton and NAP

The contributions of two separate particle populations to bulk backscattering are modeled using Mie theory (Mie, 1908) for homogeneous spherical particles and the Aden-Kerker (Aden and Kerker, 1951) method for coated spheres. Living phytoplankton

cells are represented by the first particle population, and all other suspended particles of any origin (i.e. non-algal particles, NAP) are represented by the second population. Living phytoplankton cells are modeled as coated spheres using the Equivalent Algal Populations (EAP) framework (Bernard et al., 2009; Robertson Lain et al., 2014; Robertson Lain and Bernard, 2018) for determining optical model inputs, in particular the complex indices of refraction of the particle core and coat. NAP are modeled as homogeneous spheres meant to represent organic detritus, but also allowing for their real index of refraction to vary over a

wider range to take into account the contribution of mineral particles.

A characteristic of the PSD algorithm presented here is that it is mechanistic to the extent feasible, i.e. based on first principles and causality, even at the expense of increasing complexity. For example, as in EAP, the imaginary refractive index (RI) of the cell is a function of intracellular chlorophyll concentration, $Chl_i$. Importantly, here we vary some optical model inputs in a Monte Carlo simulation in order to assess uncertainty and base the PSD inversion on an ensemble of forward runs rather than

a single specific set of inputs. Details of uncertainty estimation and propagation are given in Supplement Sec. S1. Details of how each input parameter for phytoplankton cells and for NAP is specified, as well as the statistical distributions from which the Monte Carlo simulation instances were picked are specified in Table 1 and Table 2.





As in the EAP model, the chloroplast is represented by the particle coat. Its relative volume, $V_s$, is picked from a distribution as shown in Table 1. The chloroplast's imaginary refractive index (RI) (relative to seawater) at 675 nm, $n'(675)$, is then
computed as follows (Morel and Bricaud, 1986; Bernard et al., 2009; Robertson Lain and Bernard, 2018):

$$n'(675) = \frac{Chl^* \times Chl_i \times 10^6 \times 675 \times 10^{-9}}{4\pi \times V_s \times n_{sw}(675)} \tag{2}$$

where $Chl^* = 0.027$ m$^2$mg$^{-1}$ is the theoretical maximum specific absorption coefficient of chlorophyll at 675 nm when dissolved in water (Bernard et al., 2009; Robertson Lain and Bernard, 2018), $Chl_i$ is the intracellular chlorophyll concentration in kg Chl m$^{-3}$ of cellular material, and $n_{sw}(675)$ is seawater's absolute real RI at 675 nm. A hyperspectral basis vector from
the EAP model (based on measurements, for details see Bernard et al. (2009); Robertson Lain and Bernard (2018)) is then scaled using the value at 675 from Eq. 2, obtaining a hyperspectral relative imaginary RI for the coat as chloroplast. In Eq. 2, $Chl_i$ applies to the whole cell and is therefore scaled using $V_s$ to obtain $n'(675)$ for the coat alone. The nominal chloroplast's real relative RI is then picked from a distribution as shown in Table 1, and modified as a function of its imaginary RI according to the Kramers-Kronig relations (implemented as a Hilbert transform) (Bernard et al., 2009; Robertson Lain and Bernard,
125  2018).

The cell cytoplasm is represented by the particle core. It's real relative RI is picked from a distribution given in Table 1, and it is modified by the Kramers-Kronig relations using a constant hyperspectral detritus-like imaginary RI, i.e. having a colored dissolved organic matter (CDOM)-like exponential spectral shape, resulting in spectrally-varying hyperspectral relative real RI. The phytoplankton particle population relative RIs and their Monte Carlo variability are summarized in Supplement Fig.
S1.

The NAP population is represented by a homogeneous sphere, the relative RIs of which are picked so that its absorption spectrum is detritus-like (same as the core of phytoplankton), and its real RI is allowed to vary over a wider range of values, meant to represent mostly organic detritus, but with some minerogenic contributions, resulting in a mean nominal real relative RI of $\approx 1.06$. The input RIs and other input parameters for NAP are summarized in Table 2.

Specification of the input PSD parameters and the relationship of NAP to phytoplankton PSDs is key to the construction of the forward and inverse models. Necessarily, some key simplifying assumptions are made here in order to construct an algorithm with operational application to modern multi-spectral ocean color sensors. The two key assumptions are: 1) Phytoplankton and NAP have a power-law PSD (Eq. 1) with the same slope $\xi$, and 2) The scaling parameter $N_0$ for NAP is twice that of $N_0$ for phytoplankton (the forward model uses default values as in Tables 1 and 2. The latter assumption is chosen so
that it results in a phyto C:POC ratio of 1:3 (see Kostadinov et al. (2016a) and Behrenfeld et al. (2005), and Sec. 3.4 here) (as long as they are both estimated using the same size ranges). Together, these assumptions allow for the retrieval of one common PSD parameter set pertaining to the total particle population PSD (one $\xi$ value and one total $N_0$ equal to the linear sum of the NAP and phytoplankton $N_0$ values).



| Input Parameter | Symbol, Units & Notes |
|---|---|
| Pure Seawater Absolute Real RI | $n_{sw} = f(\lambda)$, after Zhang et al. (2009) using Temperature = 15 ° C & Salinity = 33 |
| PSD slope | $\xi \in [2.5, 6]$ in steps of 0.05. The same value applies to both phytoplankton and NAP. |
| Wavelengths *in vacuo* | $\lambda \in [400, 700]$ nm; hyperspectral - in steps of 1 nm. Band-averaging used for the nominal wavelengths of satellite sensors. |
| **Phytoplankton Population Inputs** | |
| Intracellular Chlorophyll Concentration | $Chl_i \in [0.5, 10]$, picked from, $\mathcal{N}(2.5, 2.5)$; $\mu \approx 3.14$ kg Chl m$^{-3}$ cellular material. |
| Coat (Chloroplast) Relative Volume | $V_s \in [5, 35]$ %, picked from $\mathcal{N}(20,5)$, resulting in mean coat relative thickness as fraction of cell radius $t_{coat} = 7.2\%$ (cf. Organelli et al. (2018)); $t_{coat} = 1 - (1 - V_s)^{1/3}$ |
| Coat (Chloroplast) Relative Real RI | $n_{coat} \in [1.06, 1.22]$, picked from, $\mathcal{N}(1.14, 0.08)$; $\mu \approx 1.14$; wavelength-dependent via Kramers-Kronig relations. |
| Core (cytoplasm) Relative Real RI | $n_{core} \in [1.01, 1.03]$, picked from, $\mathcal{N}(1.02, 0.01)$; $\mu \approx 1.02$; wavelength-dependent via Kramers-Kronig relations. |
| Coat (Chloroplast) Relative Imaginary RI | $n'_{coat}(\lambda)$ is computed from a hyperspectral basis vector (from Bernard et al. (2009); Robertson Lain and Bernard (2018)) that is scaled to the value at $n'(675)$ using Eq. 2. |
| Core (cytoplasm) Relative Imaginary RI | $n'_{core}(\lambda)$ has a prescribed constant magnitude & detritus-like (exponential) spectral shape, with spectral slope $S = 0.0123$ nm$^{-1}$, resulting in $S_a \approx 0.014$ nm$^{-1}$ for $a_{core}(\lambda)$. |
| Minimum outer particle diameter | $D_{min_\phi} = 0.5$ µm |
| Maximum outer particle diameter | $D_{max_\phi} \in [20, 200]$ µm, picked from, $\mathcal{N}(50,50)$; $\mu \approx 72.3$ µm; |
| Differential Number Concentration at $D_0 = 2$ µm | $N_{0\phi} = 5 \times 10^{16}$ m$^{-4}$. Used in the forward modeling. |

**Table 1.** Inputs for the coated spheres Aden-Kerker optical scattering computations for the phytoplankton particle population. Modeling inputs common to both phytoplankton and NAP (see Table 2) are given in the first three table rows. $\mathcal{N}(\mu, \sigma)$ stands for a normal distribution with mean $\mu$ and standard deviation $\sigma$.

| NAP Population Inputs | |
| --- | --- |
| Relative Real RI | $n_{NAP} \in$ [1.01, 1.2], picked from, $\mathcal{N}(1.02, 0.06)$; $\mu \approx 1.06$; wavelength-dependent via Kramers-Kronig relations. |
| Relative Imaginary RI | $n'_{NAP}(\lambda)$ has a prescribed constant magnitude & detritus-like (exponential) spectral shape, with spectral slope $S = 0.0123$ nm$^{-1}$ , resulting in $S_a \approx$ 0.014 nm$^{-1}$ for $a_{NAP}(\lambda)$. |
| Minimum outer particle diameter | $D_{min\,NAP} = 0.01$ µm |
| Maximum outer particle diameter | $D_{max\,NAP} \in$ [200, 500] µm, picked from, $\mathcal{N}(400, 10)$; $\mu \approx 376.8$ µm; |
| Differential Number Concentration at $D_0 = 2$ µm | $N_{0\,NAP} = 2 \times N_{0\phi} = 1.0 \times 10^{17}$ m$^{-4}$, resulting in total particle population $N_0 = 1.5 \times 10^{17}$ m$^{-4}$. Used in the forward modeling. |

**Table 2.** Inputs for the homogeneous spheres Mie scattering code for the NAP population. Modeling inputs common to both phytoplankton and NAP are given in the first three table rows of Table 1. $\mathcal{N}(\mu, \sigma)$ stands for a normal distribution with mean $\mu$ and standard deviation $\sigma$.

## 2.2 Backscattering Calculations

The backscattering efficiencies $Q_{bb}(\lambda)$, for a single phytoplankton cell and NAP particle were computed using using the inputs described above in Sec. 2.1 and Tables 1 and 2. The coated spheres code of Zhang et al. (2002) was used for both coated and homogeneous spheres. This code is included with the algorithm development scientific code of the PSD algorithm (see Sec. 4). Calculations were run for N = 3000 instances of Monte Carlo simulations, each with a unique randomly picked combination of inputs for phytoplankton and NAP. This resulted in 3000 sets of hyperspectral $Q_{bb}$ values. High sampling resolution in diameter

space was picked for the coated spheres (10000 samples between minimum and maximum diameter) in order to minimize the influence of resonance spikes in $Q_{bb}$. For NAP, 1000 samples of $D$ were used.

Indices of refraction for both phytoplankton and NAP are specified hyperspectrally (Supplement Fig. S1) and the computations are performed from 400 nm to 700 nm wavelength *in vacuo* with a step of 1 nm, allowing the resulting hyperspectral $Q_{bb}(\lambda)$ values to be adapted for use with any combination of visible optical wavebands pertaining to recent and currently

operating ocean color multispectral sensors, or for planned (e.g. PACE (Werdell et al., 2019)) or existing hyperspectral sensors.

Before $b_{bp}$ calculation, hyperspectral backscattering efficiencies, $Q_{bb}$, for each Monte Carlo run were first pre-processed by applying quality control, and band-averaging using a moving average 11-nm-wide top-hat filter, using as central wavelengths the nominal bands of the following ocean color sensors: Sea-viewing Wide Field-of-view Sensor (SeaWiFS), Moderate Resolution Imaging Spectroradiometer (MODIS) *Aqua*, Medium Resolution Imaging Spectrometer (MERIS) and Ocean and Land

Colour Instrument (OLCI), Visible and Infrared Imager/Radiometer Suite (VIIRS) on the Suomi National Polar-orbiting Partnership (S-NPP), plus 440 and 550 nm, resulting in 19 unique bands for band-averaged backscattering efficiencies, denoted



here as $\overline{Q_{bb}(\lambda)}$. The band-averaged spectral particulate backscattering coefficient, $b_{bp}(\overline{\lambda})$ was then calculated from the $\overline{Q_{bb}(\lambda)}$ values and the input PSD as follows (e.g. van de Hulst (1981); Kostadinov et al. (2009)):

$$b_{bp}(\overline{\lambda}) = \int_{D_{\min}}^{D_{\max}} \frac{\pi}{4} D^2 \overline{Q_{bb}}(D, \overline{\lambda}, m) N_0 \left( \frac{D}{D_0} \right)^{-\xi} \mathrm{d}D \tag{3}$$

where $m$ is the complex index of refraction (specified separately for coat and core in the case of phytoplankton). Equation 3 is applied separately to the phytoplankton and NAP modeled $Q_{bb}$ values, and for each of the 3000 Monte Carlo runs. Band-averaged total $b_{bp}(\overline{\lambda})$ spectra are then calculated as the linear sum of phytoplankton and NAP backscattering. Other IOPs can be calculated using Eq. 3 by substituting the backscattering efficiency with the corresponding efficiency for the IOP, e.g. absorption.

## 2.3   PSD retrieval via Spectral Angle Mapping

### 2.3.1   End-member construction

Band-averaged total $b_{bp}(\overline{\lambda})$ spectra were used to construct the backscattering *end-members*, $E(\overline{\lambda})$, corresponding to a specific value of the PSD slope $\xi$. First, individual total $b_{bp}$ spectra from each Monte Carlo run (N= 3000) were normalized by the value at 555 nm. The median of all normalized spectra at each waveband was used as the end-member for each PSD slope,

from $\xi$=2.5 to $\xi = 6$ in steps of 0.05 (see Table 1). This approach allows the isolation of $b_{bp}$ spectral shape (dependent on $\xi$), and spectral magnitude (dependent on $N_0$) (Eq. 3). Using the hyperspectral underlying $Q_{bb}$ values, end-members can be constructed for any desired set of wavelengths.

### 2.3.2   PSD parameter retrieval and operational application to OC-CCI ocean color data

The PSD parameters $\xi$ and $N_0$ are retrieved using the backscattering end-members, $E(\overline{\lambda})$, via the spectral angle mapping

(SAM) technique (e.g. Dennison et al. (2004)). Briefly, the end-members and satellite-observed $b_{bp}$ spectra are treated as $n$-dimensional vectors where $n$ is the number of bands. The spectral angle between a given end-member and the observed spectrum is then calculated using the vector dot product as:

$$\Theta = cos^{-1} \left( \frac{\boldsymbol{b_{bp}}(\overline{\boldsymbol{\lambda}}) \cdot \boldsymbol{E}(\overline{\boldsymbol{\lambda}})}{||\boldsymbol{b_{bp}}(\overline{\boldsymbol{\lambda}})|| \, ||\boldsymbol{E}(\overline{\boldsymbol{\lambda}})||} \right) \tag{4}$$

Thus, spectral angle is an index of spectral shape similarity between two spectra, with more similar spectral shapes resulting

in lower spectral angles. Equation 4 was used to calculate the spectral angle $\Theta$ between each of the 71 end-members,$E(\overline{\lambda})$, and the input observed $b_{bp(\overline{\lambda})}$ spectrum. The value of $\xi$ corresponding to the smallest spectral angle is then assigned as the retrieved PSD slope. Three wavebands were used, namely 490, 510 and 550 nm. For operational application to OC-CCI v5.0 (Sathyendranath et al., 2021) data (which does not have the 550 nm band), band-shifting was applied to the input $R_{rs}(560)$ to





estimate the corresponding $R_{rs}(550)$, which is used in the Loisel and Stramski (2000) IOP inversion. The band-shifting was

constructed using the band ratios between the respective original and target bands from a hyperspectral run of the Morel and

Maritorena (2001) (MM01) model. No other bands were shifted.

The $N_0$ parameter is subsequently retrieved as the ratio of 1) the satellite observed value of $b_{bp}(443)$ and 2) the median value

of the quantity $b_{bp}(443)/N_0$ corresponding to the end-member class of the retrieved $\xi$ and all statistically similar classes (see

Supplement Sec. S1) across all Monte Carlo simulations.

## 2.4   Derived products: Size-partitioned phytoplankton carbon, PSCs, POC and Chlorophyll

Once the PSD parameters are known, they can be used to compute derived products (Kostadinov et al., 2010, 2016a; Roy

et al., 2017)). Phytoplankton carbon in any size class spanning from cell diameter $D_{min}$ to cell diameter $D_{max}$ (in m) can be

estimated as:

$$\text{phyto C} = \int_{D_{\min_\phi}}^{D_{\max_\phi}} 10^{-9} a \left( 10^{18} \frac{\pi}{6} D^3 \right)^b N_{0\phi} \left( \frac{D}{D_0} \right)^{-\xi} dD \tag{5}$$

where $N_{0\phi} = \frac{1}{3} N_0$ , and $N_0$ (m$^{-4}$) for the total PSD is the satellite-retrieved parameter from total particulate backscattering;

the other PSD parameters are as in Eq. 1. Equation 5 was used to compute size-partitioned phyto C in three size classes -

picophytoplankton (0.2 to 2 μm in diameter), nanophytoplankton (2 to 20 μm in diameter) and microphytoplankton (20 to 50

μm in diameter), as well as total phyto C as the sum of the three classes. Carbon-based PSCs are defined as the fractional

contribution of each of the three size classes to total phyto C (Kostadinov et al., 2016a). Given the first-order correspondence

between PSCs and PFTs (e.g. Quéré et al. (2005)), these PSCs can also be interpreted as PFTs. The allometric coefficients

of Roy et al. (2017) are used here, namely $a = 0.54$ and $b = 0.85$; when cell volume $V$ is expressed in μm$^3$, cellular carbon

is computed in pg C per cell using these coefficients (Eq. 5, see also Menden-Deuer and Lessard (2000)). Phyto C in Eq. 5

is given in mg m$^{-3}$; the conversion factors in Eq. 5 are used to convert from m$^3$ to μm$^3$, and from pg to mg C (Kostadinov

et al. (2016a); Roy et al. (2017)). The factor of $\frac{1}{3}$ is an assumption of the model (Tables 1 and 2). Thus, an estimate of POC

(computed using the same size limits as total phyto C) was calculated as $3 \times$ phyto C.

Chlorophyll concentration was estimated from the PSD retrievals and the input intracellular chlorophyll concentration, $Chl_i$

(Table 1; Roy et al. (2017)) as follows:

$$\text{Chl} = \int_{D_{\min_\phi}}^{D_{\max_\phi}} \frac{\pi}{6} D^3 Chl_i N_{0\phi} \left( \frac{D}{D_0} \right)^{-\xi} dD \tag{6}$$

Here, $Chl_i$, $D$, $D_0$ and $N_{0\phi}$ all have to be expressed in consistent units so that Chl is obtained in mg m$^{-3}$. Here we use the

median $Chl_i$ across all Monte Carlo simulations to produce a single Chl estimate.



## 2.5 Validation and Comparison

A data set of near-surface *in-situ* PSD measurements was compiled for validation of the PSD parameter products, $\xi$ and $N_0$ (Eq. 1). The data set consists of mostly Coulter counter measurements, some LISST measurements, and a small set of PSDs derived from multiple instruments and modeling. Specifically, the compilation consists of the following data sets: 1) a compilation of

several data sets of Coulter counter measurements, as used in the KSM09 algorithm validation in Kostadinov et al. (2009), 2) LISST-100X (Sequoia Scientific©) measurements from the Plumes and Blooms Project (e.g. Toole and Siegel (2001); Kostadinov et al. (2007)) in the Santa Barbara Channel, as used in Kostadinov et al. (2012), 3) Coulter counter measurements from the Atlantic Meridional Transect #26 (AMT26) (Organelli and Dall'Olmo, 2018), as compiled and used in Organelli et al. (2018, 2020); and 4) PSDs obtained using a VSF-inversion technique (Zhang et al., 2011, 2012) from the volume scattering

functions (VSFs) measured during the NASA EXPORTS campaign (Siegel et al., 2016) in the North Pacific in 2018 (Siegel et al., 2021).

The compiled PSD data set was used to fit for the PSD parameters of Eq. 1 using the 2 to 20 μm diameter range. One data point was removed from the 2018 EXPORTS PSD data due to a poor fit to a power-law PSD. These *in-situ* estimates were matched to satellite OC-CCI v5.0 (Sathyendranath et al., 2019, 2021) satellite $R_{rs}$ using the same matching methods described

below for POC and pico-phytoplankton carbon data. Matched reflectances were used as input to the novel PSD algorithm presented here. The *in-situ* and satellite PSD parameters were then compared using a type II linear regression and several additional algorithm performance metrics (e.g. Seegers et al. (2018)), details of which are given in the Fig. 8 caption.

A large compilation of *in-situ* POC data was collected from various public databases and private contributors and was used here to perform match-ups with satellite OC-CCI v5.0 data. In addition to the POC data (1997-2012) used in Evers-King et al.

(2017) for algorithm validation (N = 3891), this study also incorporated recent *in-situ* POC data (2013-2020) from the SeaWiFS Bio-optical Archive and Storage System archive (https://seabass.gsfc.nasa.gov/). The daily, 4 km, sinusoidal projection OC-CCI v5.0 data (1997-2020) (Sathyendranath et al., 2019, 2021) were used to extract the closest central satellite pixels to the *in-situ* data points. If the central satellite match-up pixels was valid, the surrounding eight pixels (a 3 x 3 pixel box) were also extracted to estimate the mean, median, and standard deviation of all OC-CCI variables. The match-up data points were then

averaged with respect to depth (0 to 10 m), location, and date. Moreover, a number of uncertain match-up data points (e.g. with less than 4 valid pixels out of 9) were removed. A total number of 6041 match-up data points were obtained and used for analysis. Here, the median satellite $R_{rs}(\lambda)$ matched-up variables were used to compute the satellite-retrieved POC data using the PSD-based algorithm. Duplicate in-situ match-ups (in the sense of points receiving the same satellite match-up) were treated as separate match-up points.

The *in-situ* pico-phytoplankton carbon data set compiled and used for algorithm inter-comparison as part of the ESA POCO project (Martínez-Vicente et al., 2017) was used here to generate match-ups with satellite OC-CCI v5.0 $R_{rs}$ data for further validation. Match-ups were generated in the same way as described above for POC. All *in-situ* data described above were excluded from the validation if any of the following conditions were met: 1) average bathymetric depth from an $\approx 9$ km buffer around the *in-situ* sample location was less than or equal to 200 m, or any grid cell elevations in that buffer were 0 m or higher,




using a downsampled, 4 km version of the NOAA ETOPO1 data set (https://www.ngdc.noaa.gov/mgg/global/); 2) the *in-situ* sample depth was 15 m or greater, or 3) there were four or fewer satellite pixels available to use in the match-up, as detailed above.

In addition to validation against *in-situ* measurements of the PSD, POC and pico-phytoplankton carbon, satellite chlorophyll-a (Chl) retrievals (using the standard algorithm of OC-CCI v5.0 at the match-up points were compared with Chl estimated

using the EAP-based formulation of the PSD algorithm developed here and the retrieved PSD (Eq. 6). Finally, global algorithm retrievals of total phyto C for May 2015 (using OC-CCI v5.0 data as input) were also compared with two alternative methods of retrieving phyto C: 1) the Roy et al. (2017) algorithm, and 2) the Graff et al. (2012, 2015) algorithm, as implemented by NASA's Ocean Biology processing Group (OBPG). Modeling and processing of results presented here is done using the sinusoidal projection images; maps presented here are given in equidistant cylindrical projection (i.e. un-projected latitude/longitude).

## 260 3 Results and Discussion

### 3.1 Forward and Inverse Modeling

The first step in the algorithm development is the generation of 3000 Monte Carlo realizations of backscattering efficiencies as a function of particle diameter and wavelength, $Q_{bb}(D, \lambda)$. The important differences between backscattering efficiencies of homogeneous and coated particles is discussed in Supplement Sec. S2 and illustrated in Supplement Fig. S2. Here, we continue

the discussion with the resulting integrated backscattering spectra (Eq. 3. Hyperspectral $b_{bp}(\lambda)$ spectra modeled using a single forward optical model run are shown in Fig. 1. The computations use the median values of inputs that are varied in the Monte Carlo simulations (Tables 1 and 2). These normalized spectra illustrate the strong spectral shape dependence on the PSD slope $\xi$. Phytoplankton $b_{bp}$ spectral shapes are complex, with various peaks and troughs near the absorption peaks of chlorophyll, but are more linear in the 490 to 550 nm range, which is the one used for the multi-spectral operational PSD algorithm. Regardless,

the SAM methodology of retrieval here allows for any spectral shape and does not impose a power-law fit to the shape of $b_{bp}$, as is done in KSM09 (Kostadinov et al. (2009)) (see also Loisel et al. (2006)). NAP backscattering exhibits smooth shapes due to the smooth shape of their absorption (Fig. 1B). Fundamentally, it is evident from Figs. 1A and 1B that the higher the PSD slope $\xi$, the steeper $b_{bp}$ spectral shape becomes, with higher values in the blue, since smaller particles dominate the signal. This dependence is at the root of the principle of operation of the PSD algorithm. For completeness, corresponding absorption

spectra are illustrated in Supplement Fig. S3.

The 71 end-members (EMs) created for operational application to existing major satellite ocean color missions and corresponding to PSD slope values between 2.5 and 6.0 with a step of 0.05 are displayed in Fig. 2A. They represent the modeled $b_{bp}(\overline{\lambda})$ spectra against which satellite-measured $b_{bp}$ spectra are compared using the SAM method (Eq. 4). The spectral shape dependence on $\xi$ demonstrates the ability to retrieve this parameter from space.

An important question in bio-optical oceanography is determining the sources of backscattering in the ocean and their relative contributions. This is still not a resolved issue (Stramski et al., 2004), though progress has been made (e.g. Organelli et al. (2018); Koestner et al. (2020); Zhang et al. (2020)). This issue is of central importance to the PSD model, as it assigns varying




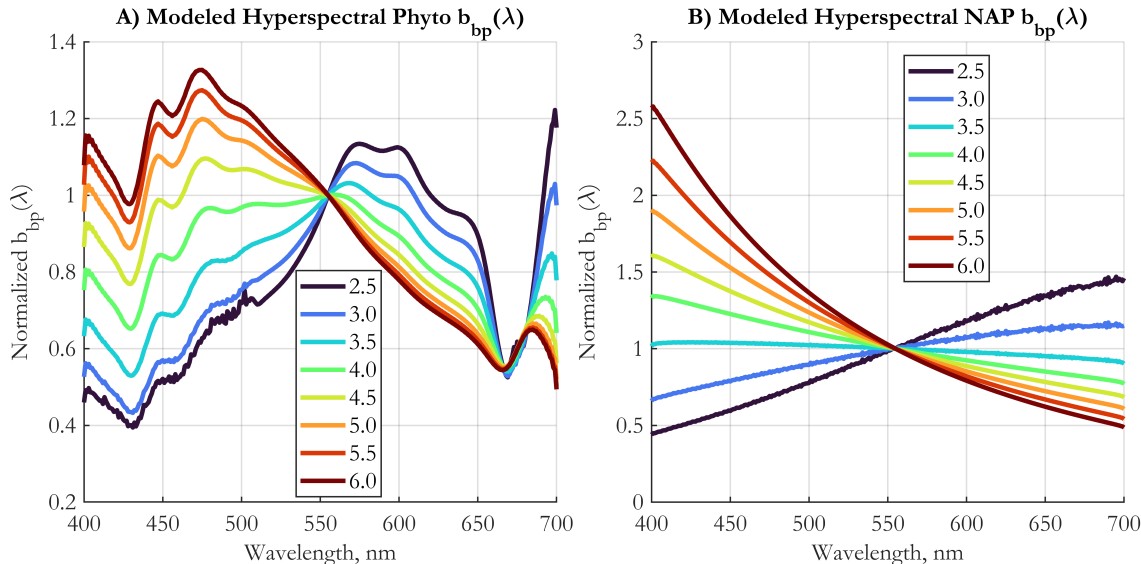

**Figure 1.** Modeled hyperspectral backscattering coefficient by (A) phytoplankton, using EAP-based coated spheres Mie scattering computations, and (B) NAP, modeled as homogeneous spheres, as a function of the input power-law PSD slope (color-coded solid lines, as in legend). All spectra are shown normalized to the respective values at 555 nm. See Sec. 2 for more details.

fractions of the $b_{bp}$ signal to phytoplankton vs. NAP, under certain assumptions (Tables 1 and 2). Since in the 2-component PSD model presented here phytoplankton and NAP $b_{bp}$ are modeled separately, the fraction of $b_{bp}$ due to phytoplankton vs. NAP

can be calculated. For a given PSD slope $\xi$ and wavelength, the assumptions of the model dictate fixed fractional contributions by NAP and phytoplankton to total $b_{bp}$, which are given in Fig. 2B. There is variability by wavelength, but the first-order variability is driven by the PSD slope, namely at low $\xi$ values ($\xi < 4.0$), phytoplankton contribution to $b_{bp}$ is on the order of 30 to 50%, and it drops off to near 0% for higher slopes as $\xi$ approaches 6.0. The curves are not monotonic, and peak phytoplankton contribution to $b_{bp}$ occurs at $\xi \approx 3.25$.

The fractional contributions of Fig. 2B are derived from the forward theoretical modeling, and they are influenced by all model assumptions and are not validated independently. In particular, the decisions of integration diameter limits for NAP and phytoplankton, as well as on the distributions of the indices of refraction for phytoplankton and NAP will have a strong influence on these values. Since NAP are here permitted to have higher RIs than RIs typical of organic detritus only, if NAP were strongly dominated by or composed only of organic particles, then NAP contribution to $b_{bp}$ would be overestimated

here. Of course these RIs are likely to be spatially and temporally variable, and the algorithm can be further improved by investigating and implementing such variability. Bellacicco et al. (2018) estimated global absolute $b_{bp}$ due to NAP and its fractional contribution to total $b_{bp}$ using analysis of correlations with Chl. Qualitatively and to first order, their global pattern of percent $b_{bp}$ due to NAP agrees with the model results reported here, i.e. low relative NAP contributions in high latitudes and eutrophic areas, and higher relative contributions in more oligotrophic areas such as the fringes of the subtropical gyres

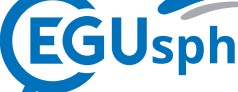



**Figure 2.** (A) Normalized spectral shapes of the $b_{bp}$ end-members developed for spectral angle mapping (SAM), shown at the 19 unique wavelengths used for band-averaging (See Sec. 2.2). (B) The fraction of $b_{bp}(\lambda)$ due to phytoplankton as a function of the PSD slope $\xi$. The wavelengths shown are indicated in the legend (nm). The means across all 3000 Monte Carlo simulations are shown. (C) Uncertainties in the PSD slope $\xi$ retrieval using the SAM method, for each end-member (EM). Shown are the minimum and maximum value of the PSD slope for all end-member classes that are statistically similar to the given EM, according to the Kruskal-Wallis ANOVA (Supplement Sec. S1) (left y-axis), and the resulting range of PSD slopes (right y-axis) falling within these asymmetric uncertainty bounds. (D) Statistics of the parameter $log10(b_{bp}(443)/N_0)$ for each EM, calculated for all 3000 Monte Carlo simulations and across all neighboring EM classes determined to be statistically similar to the given EM. $\mu$ in the legend stands for the mean, and $\sigma$ - the standard deviation. The standard deviation of this parameter is used to estimate uncertainties in the $N_0$ retrieval.





(they exclude the gyres from their analysis) (cf. their Fig. 2C and Fig. 2B here). Note that the Bellacicco et al. (2018) estimate pertains only to NAP non-covarying with Chl, making comparison harder. Further investigation is warranted to more rigorously compare their product to the values implicit in the PSD algorithm described here. Apart from analyzing the relative contribution of phytoplankton vs. NAP to total $b_{bp}$, it is of interest to investigate the relative contributions of various size ranges to the modeled backscattering coefficient. This is illustrated in Supplement Fig. S4 and further discussed in Supplement Sec. S3.

The uncertainty in PSD slope $\xi$ retrieval as a function of $\xi$ is illustrated in Fig. 2C. These estimates are not symmetric about the $\xi$ value and are derived via Kruskal-Wallis analysis of variance to determine class similarity (Supplement Sec. S1). As in KSM09, the general tendency is for the range of uncertainty in $\xi$ to increase for lower PSD slopes, but it is always less than 0.5. The uncertainty in the $b_{bp}(443)/N_0$ ratio used to retrieve the $N_0$ parameter is shown in Fig. 2D in log10 space. Mean and median values are similar, and the uncertainty about them does not vary much with PSD slope, also similarly to KSM09. 310 Importantly, the uncertainty in Fig. 2D at each $\xi$ value includes all statistically similar classes of EMs.

### 3.2 Operational Application of the PSD/Phyto C Algorithm to OC-CCI v5.0 Merged Satellite Data

#### 3.2.1 PSD Parameters

The operational PSD algorithm presented here was applied to the monthly 4-km OC-CCI v5.0 $R_{rs}(\lambda)$ data set (Sathyendranath et al. (2019, 2021)). Both PSD parameters ($\xi$ and $N_0$, Eq. 1) and derived products were generated (Sec. 2.3 and Sec. 2.4). These 315 data and their monthly and overall climatologies (and associated uncertainties) are made publicly available (see 4). Here, we use May 2015 data to illustrate and discuss the new algorithm.

The PSD map (Fig. 3A) reveals a global spatial pattern consistent with expectations and with KSM09, namely the subtropical oligotrophic gyres are characterized by high PSD slopes, i.e. relatively high numerical dominance of small particles, whereas more eutrophic areas such as coastal areas, Equatorial upwelling zones, and high latitudes exhibit lower slopes, i.e. increasing 320 relative abundance of larger particles. This is consistent with oligotrophic ocean ecosystems being dominated by picophytoplankton, whereas microphytoplankton contribute significantly to the phytoplankton assemblage in eutrophic areas and during blooms (e.g. Kostadinov et al. (2009, 2010)). PSD slope values retrieved by the SAM-based algorithm span the full modeled range of $2.5 \leq \xi \leq 6.0$. This is in contrast to KSM09, where values below 3.0 were not retrieved. The $N_0$ PSD parameter (Eq. 1) is, as expected, higher in coastal, high latitude and eutrophic areas (indicating higher particle loads), and lower in the 325 oligotrophic subtropical gyres (Fig. 3B). $N_0$ varies over a few orders of magnitude, and it is generally the first order control on absolute particle loads in seawater. Note $N_0$'s units of $\mathrm{m}^{-4}$ (Eq. 1) and that care should be taken when comparing Eq. 1 and $N_0$ to other formulations of the PSD, e.g. the $k$ parameter in Roy et al. (2017), as these are related, but not equivalent (see also Vidondo et al. (1997)).

Algorithm uncertainties are provided on a per-pixel basis. The uncertainty range estimates for $\xi$ (Fig. 3C) (not necessarily 330 symmetric about the $\xi$ value) indicate that the gyres are characterized by lower uncertainties than the more eutrophic areas, as can be expected from Fig. 2C. These are partial uncertainty estimates, including those quantifiable and internal to the modeling, i.e. due to Mie parameter choices. Additional uncertainties inherent in the input OC-CCI $R_{rs}$ values and those due to the IOP




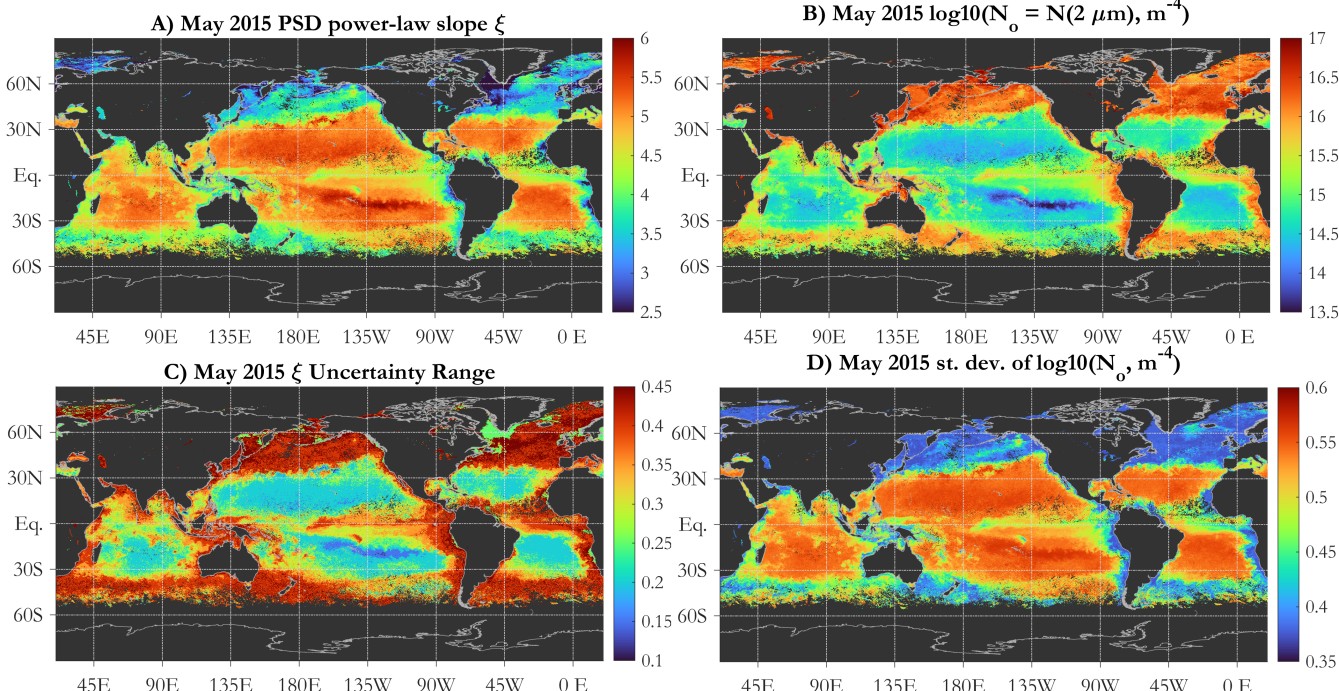

**Figure 3.** Example operational retrievals of the PSD parameters (Eq. 1) and their uncertainties, using monthly OC-CCI v5.0 $R_{rs}$ data for May 2015: (A) PSD slope $\xi$ (A); (B) $N_0$ parameter (in $\mathrm{m}^{-4}$ in log10 space); (C) uncertainty range for $\xi$, and (D) standard deviation of log10 of $N_0$.

inversion algorithm used are not included in Fig. 3C and 3D and in subsequent propagated errors. Uncertainties of the $N_0$ parameter are more uniform spatially, but higher in the gyres (Fig. 3D. Note that those are given in log10 space as a standard deviation, and a relatively small absolute value of the uncertainty translates to relatively large uncertainties in absolute particle concentrations.

### 3.2.2 Phytoplankton Carbon and Carbon-based PSCs; POC and Chlorophyll from the PSD

Global patterns of total phytoplankton carbon retrieved via the PSD and allometric relationships (Fig. 4A) exhibit the expected lower values in the oligotrophic gyres and higher values elsewhere. Similarly to the results of the Kostadinov et al. (2016a) algorithm, values range over approximately 3 orders of magnitude, which is a higher range than retrievals based on other methods, namely direct empirical algorithm POC retrieval (Stramski et al., 2008) or the Behrenfeld et al. (2005) method of scaling backscattering, and it is also higher than the range in CMIP5 model ensembles (cf. Fig. 1 in Kostadinov et al. (2016a)). This putative underestimation in the gyres and overestimation in eutrophic areas suggests the need for algorithm tuning, which is discussed in Sec. 3.3 along with implications of validation results. Global validation of phyto C retrievals with analytical



phyto C measurements is planned, but is currently challenging as phytoplankton-specific carbon data are relatively novel (Graff et al., 2012, 2015) and still scarce. Here, an initial validation effort is undertaken using several other variables, see Sec. 3.3.

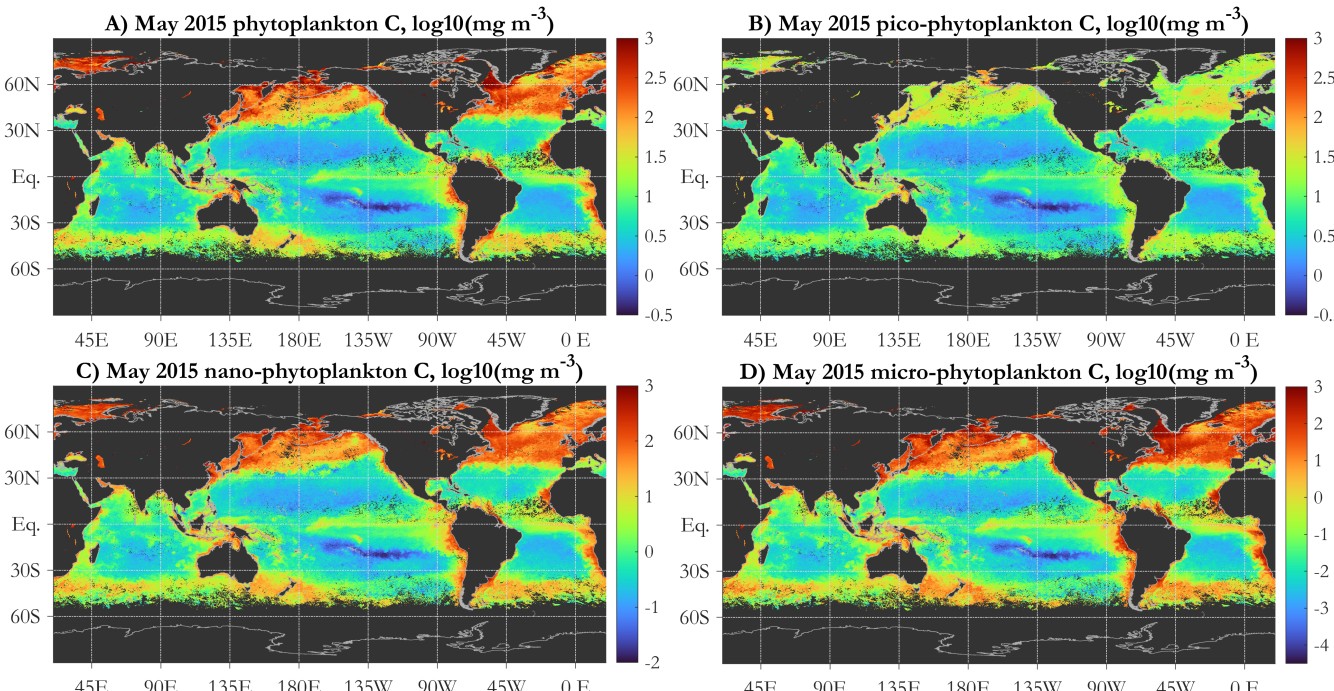

**Figure 4.** Example operational PSD-based retrievals of total size-partitioned phytoplankton carbon, using monthly OC-CCI v5.0 $R_{rs}$ data for May 2015: total phytoplankton carbon (A), pico-phytoplankton carbon (B), nano-phytoplankton carbon (C), and micro-phytoplankton carbon (D). Units are $\mathrm{mg\,m^{-3}}$, mapped in log10 space. The diameter limits for the three size classes are: picophytoplankton (0.2 to 2 μm), nanophytoplankton (2 to 20 μm) and microphytoplankton (20 to 50 μm).

A key feature of the PSD-based algorithm is that phyto C can be partitioned into any number of size classes by choosing appropriate integration limits of Eq. 5. Absolute concentrations of pico-, nano-, and micro-phytoplankton are illustrated for May 2015 in Fig. 4B, C, and D, respectively. Pico-phytoplankton C is mapped on the same color scale as total phyto C (Fig. 4A),

but pico- and nano-phytoplankton C maps have differing scales, illustrating that while pico-phytoplankton C varies over ≈ 3 orders of magnitude spatially globally, nano-phytoplankton C varies over ≈ 4-5 orders of magnitude, and micro-phytoplankton — over ≈ 7 orders of magnitude spatially (see also Kostadinov et al. (2010, 2016a). Fractional contributions of each of the three PSCs used here to total phyto C are illustrated in Fig. 5. Pico-phytoplankton dominate much of the open-ocean, lower latitude oligotrophic areas, contributing nearly 100% of the carbon biomass there (Fig. 5A), nano-phytoplankton contribute up

to ≈ 50% of biomass in the higher latitude and more eutrophic areas, and micro-phytoplankton contribute significantly only in the most eutrophic areas, e.g. during the North Atlantic bloom at ≈ 45-50° N latitude (May 2015 is shown). As previously noted (Kostadinov et al., 2010, 2016a), this general pattern is consistent with current understanding of ocean ecosystems.



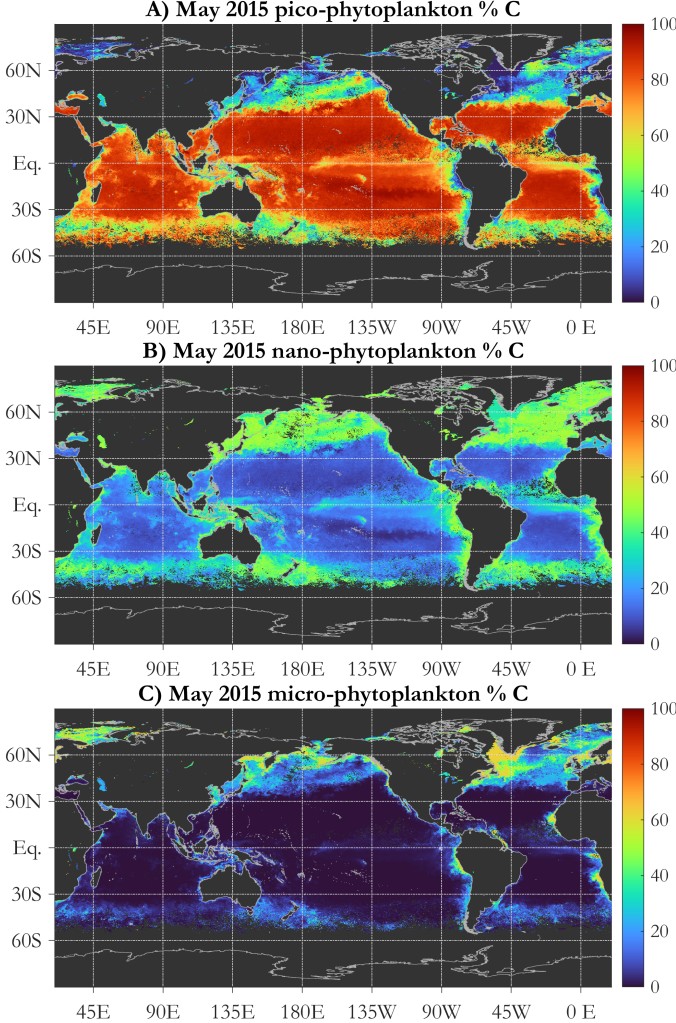

**Figure 5.** Example operational retrieval of the percent contribution of each phytoplankton size class (PSC) to total phytoplankton carbon, determined via the PSD (Sec. 2.4). Retrievals are using monthly OC-CCI v5.0 $R_{rs}$ data for May 2015. The PSCs are: pico-phytoplankton (A), nano-phytoplankton carbon (B), and micro-phytoplankton carbon (C).

The fractional carbon-based PSCs (Fig. 5) are ratios of two integrals of Eq. 5, thus they are analytical functions of the PSD slope $\xi$ and $b$, the allometric coefficient (and the limits of integration used for each class and total phyto C). The $N_0$ parameter and the $a$ allometric coefficient cancel. These functions are plotted in Fig. 6, together with the satellite-observed $\xi$ histogram for May 2015, illustrating which the most common values for the PSCs in the ocean are. Area-wise, the ocean is dominated by oligotrophic areas with high pico-phytoplankton contributions to C biomass.





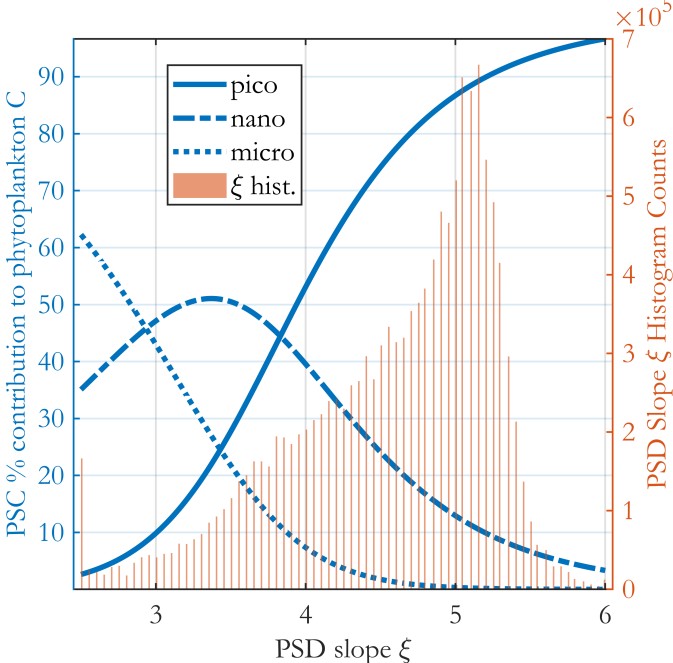

**Figure 6.** Percent contribution of each PSC to total phytoplankton carbon (blue curves as in legend, left y-axis), as a function of the PSD slope $\xi$. A histogram of the PSD slope from the (sinusoidal projection) OC-CCI v5.0-based image for May 2015 is shown in the background in red (right y-axis). The three PSC curves are analytically derived from the model, and no satellite data is used in producing them.

As an illustration of uncertainty propagation to derived products, the propagated uncertainty to total phyto C (Fig. 7A) and fractional pico-phytoplankton C biomass (Fig. 7B) are shown. Comparison of Fig. 4A with Fig. 7A indicates that absolute total

phyto C uncertainties are of the same order of magnitude as the values themselves. As noted earlier, this is a partial uncertainty estimation due to the assumed distributions of the Mie inputs (Tables 1 and 2), and due to the allometric coefficients. The Mie inputs are varied over wide ranges to accommodate various environments in the global ocean, with the goal of having a single first-principles-based operational algorithm applicable to first order globally. This increases the uncertainty estimates. The uncertainty for the fractional PSC products depends only on the uncertainties in $\xi$ and $b$. For pico-phytoplankton, they are

$< \approx 2\%$ for the oligotrophic gyres, and do not exceed $\approx 7\%$ globally. As stated above, the $N_0$ parameter and $a$ (the allometric coefficient) cancel when fractional PSCs are calculated, thus they exhibit much lower internal algorithm uncertainty compared with absolute values. This suggests that the fractional PSCs are more reliable products than the absolute values, and they can also be used with other products to partition them - e.g. total phytoplankton carbon estimated using the alternative methods shown in Fig. 9 (namely Graff et al. (2015) and Roy et al. (2017)), or the Behrenfeld et al. (2005) or Sathyendranath et al.

(2020) methods; POC products (e.g. Stramski et al. (2008)) can be partitioned this way as well. Figs. 7A and 7B illustrate some propagated uncertainties. Per-pixel uncertainties are estimated for all products and composite imagery as well (climatologies) provided with the OC-CCI-based data set associated with this manuscript (Sec. 4).



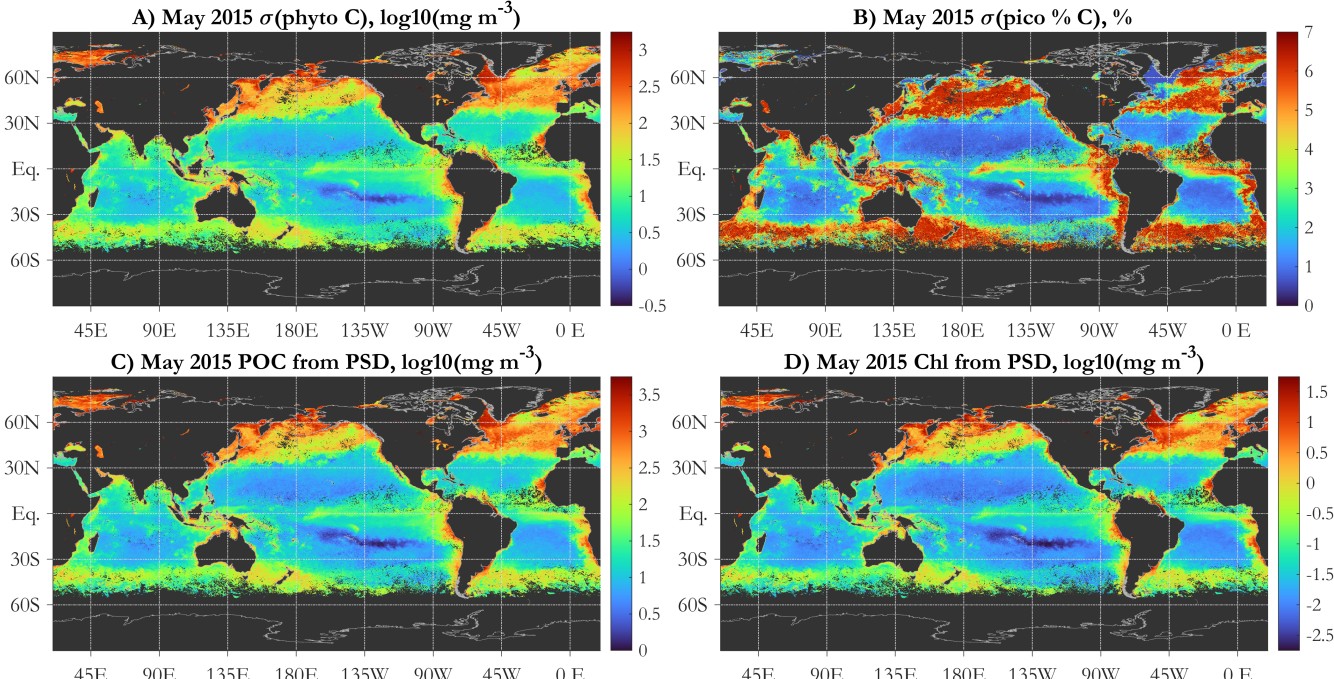

**Figure 7.** Propagated uncertainty of (A) total phytoplankton carbon given as one standard deviation (in $\mathrm{mg\,m^{-3}}$), mapped in log10 space, and (B) fractional contribution of pico-phytoplankton to total phytoplankton carbon, given as one standard deviation in percent. (C) POC derived using the PSD retrievals, (in $\mathrm{mg\,m^{-3}}$), mapped in log10 space; and, (D) Chlorophyll concentration (Chl) derived using the PSD retrievals, (in $\mathrm{mg\,m^{-3}}$), mapped in log10 space. Monthly OC-CCI v5.0-based data for May 2015 is shown in all panels.

The formulation of the PSD algorithm allows for both POC and Chl (Eq. 6) to be estimated from the retrieved PSD. Due to the assumptions used, POC is phyto C multiplied by three (Fig. 7C). This is strictly true only if the POC estimate uses the
same limits of integration as phyto C, which is an approximation to the usual POC operational definition (e.g. see discussion of POC-PSD closure analysis in Kostadinov et al. (2016a)). POC is estimated to first order, treating the retrieved NAP as being composed of POC only, and applying the same allometric relationships to it as phyto C, in spite of the fact that the assumed RI distribution of the NAP is broader (Table 2). These are simplifying assumptions of the 2-component model; a more accurate POC representation can be achieved if organic and inorganic NAP are modeled as separate particle populations (e.g.
Duforêt-Gaurier et al. (2018)). This is a planned development of the model in the future; the goal here is to build an operational PSD/phyto C algorithm for use with multi-spectral satellite data of limited degrees of freedom. Hyperspectral sensors such as PACE (Werdell et al., 2019) should allow for more degrees of freedom and thus for more independent particle components and their PSDs to be modeled separately. However, note that even hyperspectral data has limits on its degrees of freedom that are expected to be much fewer than the number of sensor bands (Lee et al., 2007; Cael et al., 2020). An important benefit of POC
is that it is a widely observed variable, available for global validation efforts (Sec. 3.3). Similarly to POC, there are benefits of the PSD-derived estimate of Chl (Fig. 7D) - it can be used as additional verification/validation of model retrievals, and/or



PSD-retrieved Chl can be used as a parameter to optimize for in algorithm tuning, as discussed shortly (Sec. 3.3). Like POC, many global *in-situ* and satellite observations of Chl are available for such efforts. Next, we discuss validation/verification and tuning efforts in which both PSD-derived POC and PSD-derived Chl are used.

## 3.3 Algorithm Validation and Empirical Tuning for the $N_0$ PSD Parameter and Absolute Concentrations

In an initial validation effort, the novel PSD/phyto C algorithm is validated/verified using several variables. It is challenging to directly globally validate the major products of the algorithm - the PSD and size-partitioned phyto C, due to a paucity of *in-situ* observations which are further reduced when performing satellite match-ups. Here, we validate or verify algorithm performance against compilations of the following variables: 1) *in-situ* PSD observations (Sect 2.5); 2) *in-situ* POC observations; 3) *in-situ*

pico-phytoplankton C observations; 4) concurrent satellite observations of Chl. Maps of the locations of *in-situ* observations are shown in Supplement Fig. S5. In addition, we compare phyto C retrievals to several existing methods using the example May 2015 OC-CCI v5.0 image. Below we discuss the results of these validation, verification and comparison efforts and a suggested tuning of the algorithm.

Validation results for the PSD slope $\xi$ (Fig. 8A) indicate a statistically significant but noisy relationship between retrieved

and observed slopes, with a positive bias for satellite retrievals, and two distinct clusters of points for which satellite values differ - centering about 3.25 vs. 4.75, respectively, but the *in-situ* values differ less and both cluster around 3.5 to 4.0. There is generally a clear tendency for points from more oligotrophic areas (as indicated by Chl color coding) to exhibit higher satellite values, and more eutrophic areas to exhibit much lower satellite values. This tendency is weaker for the *in-situ* observations, which tend to have a narrower range, mostly between 3.0 and 4.5. The same validation regression with the points classified

by location, rather than Chl, is shown in Fig. 8C. Data from two specific locations are numerous and also likely drive the regression to a large degree - 1) the Plumes and Blooms (PnB) Project LISST data, which represents a coastal site, namely the Santa Barbara Channel in California, USA (SBC), and 2) the Equatorial Pacific (EqPac). While at SBC the satellite data underestimates PSD slopes, it overestimates them at EqPac. Data from higher latitudes from various locations in the North hemisphere, from CA coastal areas other than PnB, and from the Southern Ocean span a wider *in-situ* range, which is captured

by the satellite retrievals, albeit with substantial noise. Overall, the satellite retrievals capture the *in-situ* data trend to first order.





**Figure 8.** (A) Comparison of PSD slope derived from *in-situ* measurements with the matched-up satellite retrieval (Sec. 2.5). Points are color-coded according to the corresponding satellite OC-CCI v5.0 Chl (colormap in $\mathrm{mg\,m^{-3}}$ in log10 space). Type II regression is used, and regression and validation statistics are given in the figure panel. 'y-int' stands for the y-intercept, RMS - room mean square (square root of the mean of squared differences between the *in-situ* and satellite values), Bias is the mean of the satellite minus *in-situ* values, and MAE - mean absolute error (the mean of the absolute values of the differences between the *in-situ* and satellite values) (e.g. Seegers et al. (2018)). (B) Same as in panel A, for the $N_0$ parameter (Eq. 1) (axes in log10 space). (C) The same validation regression as in panel A, but the points are color and symbol coded according to geographic area, as follows: Plumes and Blooms project (e.g. Toole and Siegel (2001); Kostadinov et al. (2007)) (PnB, green 'x'); Equatorial Pacific (EqPac, red circles); Equatorial Indian Ocean (EqIO, red '+'); Southern Ocean (SO, black '*'); California (CA) coastal area (purple squares); higher latitude Northern Hemisphere points (> 30° latitude, NH, cyan diamonds), and South Atlantic (SA, black triangles); (D) same as in panel C, but for the $N_0$ parameter (axes in log10 space).



Validation for the $N_0$ parameter (Fig. 8B) is statistically significant but noisy, with a somewhat better $R^2$. Clustering of the *in-situ* observations around $10^{15.5}$ - $10^{16.0}$ m$^{-4}$ is observed, and the majority of these observations are underestimated in the satellite retrievals. Since $N_0$ is the PSD scaling parameter which generally controls absolute number, volume and carbon concentrations variability to first order, this has implications for the global pattern of phytoplankton carbon retrievals (Fig. 4),

namely it is consistent with underestimation in the oligotrophic gyres (mostly lower Chl) and overestimation in the eutrophic areas. Inspection of the location-coded plot (Fig. 8D) indicates that a lot of the underestimated points come from the Equatorial Pacific clustered around *in-situ* $N_0 = 10^{15.75}$ m$^{-4}$; many overestimated points are from the SBC.

This pattern of under- and overestimation in the $N_0$ validation suggests an empirical tuning to absolute phytoplankton carbon of the TK16 algorithm, via a linear (in log10 space) tuning of $N_0$, as done in TK16 (Kostadinov et al., 2016a), who based the

tuning on the validation regression. A similar approach is proposed here, but it is derived differently. Details of the tuning derivation procedure are given in Supplement Sec. S4. The following global tuning equation was obtained:

$$N_0\_tuned = 10^{0.3859 \, log10(N_0)+9.5531} \tag{7}$$

where $N_0$ is the original (un-tuned) PSD parameter. This tuning changes $N_0$ retrievals in a similar fashion to the TK16 tuning and a tuning suggested by the $N_0$ *in-situ* validation reported here (Fig. 8B), namely, low $N_0$ values are increased, and high

$N_0$ values are decreased, decreasing the overall range of variability of retrieved $N_0$ and thus the range of the retrieved derived variables as well. This addresses the low bias in oligotrophic gyres and the high bias in eutrophic areas. The goal of the tuning is to get more realistic absolute retrievals of POC and Chl (hypothesizing that this should also lead to more realistic phyto C retrievals as well - however, see discussion below about the pico-phytoplankton C validation).

The tuned $N_0$ parameter for May 2015 is mapped in Supplement Fig. S6A. The overall spatial pattern of higher values in more eutrophic areas is preserved, but the global range of values is reduced compared to the original value, increasing $N_0$ in

the gyres, and decreasing it in more productive areas. The resulting multiplicative factor to be applied in linear space to phyto C, POC and Chl values is mapped in log10 space in Supplement Fig. S6B. Values in red are greater than unity in linear space (mostly between 1 and 10), indicating that the tuning increases phyto C, POC and Chl in these areas, up to about an order of magnitude (in limited areas mostly in the South Pacific gyre), and more moderately elsewhere in the tropical and subtropical,

oligotrophic oceans. The Equatorial upwelling areas and other transitional zones are not changed, and high latitude oceans exhibit correction factors mostly less than unity in linear space (mostly between 0.1 and 1), which decreases phyto C and Chl up to an order of magnitude (rare, mostly less). This tuning is not applied to figures previously discussed here.

### 3.3.1 Comparison of the PSD-based phytoplankton carbon retrieval with existing satellite algorithms

In this section, we compare phyto C retrievals with two existing methods for its retrieval. The May 2015 original total phyto C

retrieval is compared with the tuned total phyto C and the retrievals of the absorption- and PSD-based algorithm of Roy et al. (2017) and with the Graff et al. (2015) algorithm in Supplement Fig. S7. The histograms of these four images are compared in Fig. 9. The most notable feature of these comparisons is that the tuned PSD-based retrievals are similar to those of Graff et al. (2015), whereas the original PSD-based retrievals are similar to those of Roy et al. (2017), and the latter two have exaggerated





ranges globally compared to the former two. Of these algorithms, the simplest is the Graff et al. (2015), as it is a direct scaling of
$b_{bp}$, and it is based on *in-situ* chemical analytical measurements of phyto C (Graff et al., 2012, 2015). These dichotomous inter-
comparison results suggest that further algorithm inter-comparison and validation with direct *in-situ* measurements of phyto
C are needed to guide future algorithm developments; however these data are relatively novel and scarce globally. Validation
results using *in-situ* POC and pico-phytoplankton carbon (discussed next) exhibit a similar dichotomy.

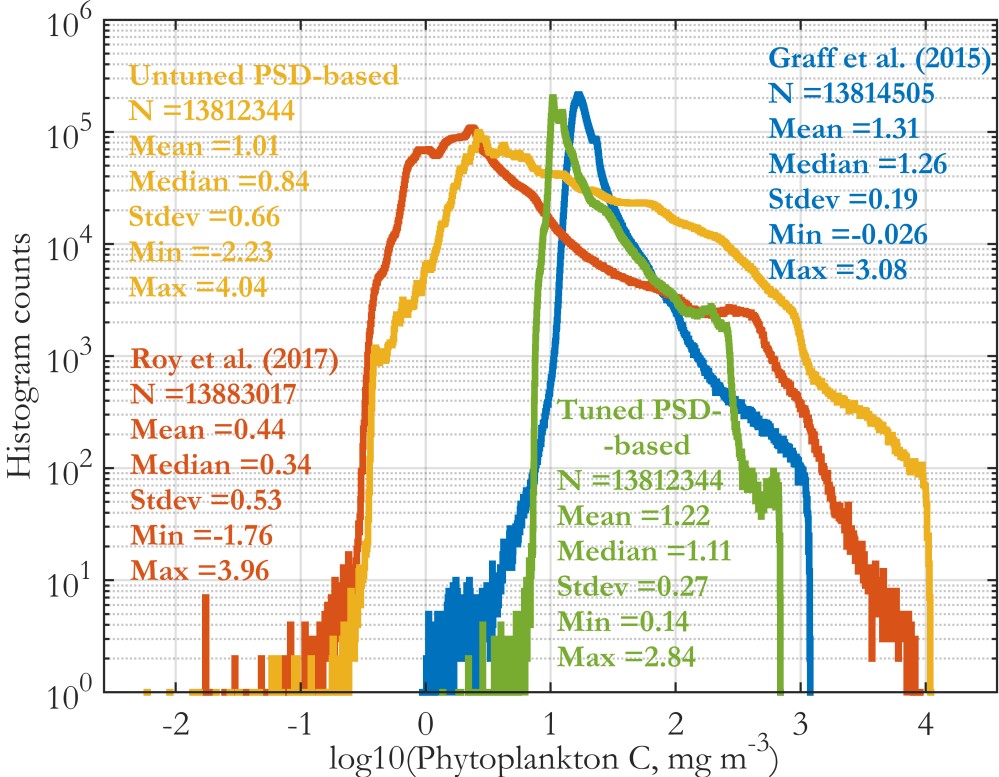

**Figure 9.** Histograms of the images of Supplement Fig. S7, including the original PSD-based phyto C retrieval (Fig. 4A). Histogram counts
are given on a log10 scale on the y axis, and the variable (x axis) is log-transformed as well. Values less than $\approx 0.3 \, \mathrm{mg \, m^{-3}}$ are exceedingly
rare for any of the retrieval variants and are shown here for completeness. All four histograms are derived from the sinusoidal projection
images for May 2015, using monthly OC-CCI v5.0 data.

### 3.3.2   Validation using POC and pico-phytoplankton carbon *in-situ* data

PSD-based estimates of POC are validated against *in-situ* measurements for the original algorithm (Fig. 10A), and the tuned
algorithm (Fig. 10C). Both regressions have satisfactory $R^2$ values, and illustrate that in general higher POC values are as-
sociated with higher Chl (colormap). Notably, the original algorithm validation has a slope of $\approx 2$ and exhibits substantial
underestimates at low POC, and overestimates at high POC. As intended, the tuning corrects this range exaggeration, and





significantly improves the slope, intercept, bias, RMS, and MAE. The regression with the $N_0$ tuning applied should not be
considered a truly independent validation, because the algorithm has been empirically tuned to retrieve POC well; the tuning
was done with global POC imagery that uses the Stramski et al. (2008) empirical POC algorithm, not with these in-situ POC
data directly.

In addition to the validation with *in-situ* POC, we performed a comparison of the matched satellite Chl and the corresponding
PSD-based Chl estimate (Eq. 6), for the original (Fig. 10B) and the tuned algorithm (Fig. 10D). Both comparisons exhibit very
high $R^2$ values, and similarly to POC, the original algorithm underestimated Chl at low values, and overestimated at high Chl
values. The tuning successfully addresses this, leading to excellent overall comparison of the tuned algorithm, with slope near
1.0, low intercept, near nil bias, and low RMS and MAE values. However, for the lowest Chl values (Chl < 0.1 $\mathrm{mg\,m^{-3}}$),
performance deteriorates. The tuned comparison is not a fully independent validation, as the algorithm was tuned to compare
well with OC-CCIv5.0 satellite retrievals (using global monthly images for 2004 and 2015). Overall, the comparison with Chl
is encouraging, indicating that the model is able to reasonably reproduce (with tuning) OC-CCI v5.0 standard satellite Chl
values at the match-up points.

Validation against *in-situ* pico-phytoplankton carbon is presented in Fig. 11A (with no $N_0$ tuning applied), and in Fig. 11C
with the tuning applied. The corresponding Chl comparisons between matched standard OC-CCIv5.0 Chl and Chl derived via
the PSD model are shown in Fig. 11B (with no $N_0$ tuning applied), and in Fig. 11D with the tuning applied. As with the POC
match-ups (Fig. 10B and D), comparisons with Chl are better for the tuned version of the algorithm, indicating that the tuning
is needed to reproduce more realistic Chl values globally. However, the tuning does not lead to any improvement in the valida-
tion results of pico-phytoplankton C (cf. Fig. 11A and C). The validation regression without tuning is statistically significant
(p<0.05), albeit noisy (low $R^2 = 0.18$); satellite retrievals and *in-situ* data cover approximately the same ranges, and increasing
Chl and *in-situ* pico-phytoplankton C generally correspond to increasing satellite values as well. However, the tuned satellite
retrievals have a very narrow range that does not cover the range of the *in-situ* data, and validation statistics are generally worse
than those of the original validation (the regression is not significant at the p=0.05 level). These validation results are generally
consistent with the results of Martínez-Vicente et al. (2017), where the tuned version of the TK16 (Kostadinov et al., 2016a)
algorithm was used.

The number of matched up sample points in the validation regression is very different among PSD, POC and pico-phytoplankton
C, and their geographic distribution is different as well. Namely, there are an order of magnitude more POC match-ups than
pico-phytoplankton carbon ones, and there are even fewer PSD points. A lot of the PSD data come from the Equatorial Pacific
and off of California, whereas POC samples are more global with a large number coming from the Atlantic Ocean, where a lot
of the pico-phytoplankton C points are located as well, and some of the latter are also from the Mediterranean. Thus the different
validation results presented here do not necessarily represent the same oceanographic conditions, e.g. the pico-phytoplankton
C *in-situ* data has less representation of eutrophic areas and spans a smaller range of Chl than the POC validation, with very
few points exceeding Chl = 1.0 $\mathrm{mg\,m^{-3}}$ (cf. Fig. 10B and 11B). Furthermore, size is not the only distinguishing characteristic
of the PSCs — differences in internal composition and structure, as well as shape, can be important too, and can significantly
affect $b_{bp}$ and the validation results.





**Figure 10.** Validation of PSD-based POC retrievals vs. *in-situ* POC measurements, and comparison of satellite retrievals of Chl using the standard OC-CCI v5.0 algorithm vs. Chl estimated from the PSD. Panels (A) and (B) have no tuning applied, whereas empirical tuning is applied to the $N_0$ parameter (Eq. 7) for panels C and D. The data points in panels A and C are colored by their matched standard OC-CCI v5.0 satellite Chl values (colormap, in $\mathrm{mg\,m^{-3}}$ in log10 space).






**Figure 11.** Validation of pico-phytoplankton carbon derived from the PSD model using daily OC-CCI v5.0 satellite data vs. *in-situ* measurements, as used in the POCO project (Martínez-Vicente et al., 2017) (panels A and C). Comparison of PSD-derived satellite Chl (y-axes) with the matched satellite retrieval of Chl using the standard OC-CCI v5.0 algorithm, at the locations of the *in-situ* pico-phytoplankton carbon match-up points (panels B and D). Panels (A) and (B) have no tuning applied, whereas empirical tuning is applied to the $N_0$ parameter (Eq. 7) for panels C and D. The data points in panels A and C are colored by their matched standard OC-CCI v5.0 satellite Chl values (colormap, in $mg\,m^{-3}$ in log10 space).



### 3.4 Further Discussion, Summary, and Conclusions

The novel PSD/phyto C algorithm described here represents a major overhaul of the KSM09 algorithm (Kostadinov et al.,
2009). Unlike KSM09, two distinct particle populations are used - phytoplankton and NAP. Phytoplankton backscattering
is modeled using coated spheres Mie calculations with inputs based on the Equivalent Algal Populations (EAP) approach
(Bernard et al., 2009; Robertson Lain and Bernard, 2018). This model formulation allows assessment of percent contribution
of phytoplankton and NAP to total $b_{bp}$, as well as Chl to be estimated from the retrieved PSD. Underlying $b_{bp}$ forward modeling
is hyperspectral, facilitating adaptation of the algorithm to upcoming hyperspectral sensors like PACE (Werdell et al., 2019).
PSD retrieval is achieved via spectral angle mapping (SAM), and no spectral shape is imposed on $b_{bp}$; operational end-members
for current and past multi-spectral sensors and the OC-CCI v5.0 merged ocean color data set are created via band-averaging
from the underlying hyperspectral modeled $b_{bp}$. The algorithm has been used to create an accompanying data set based on the
OC-CCI v5.0 data set (See Sec. 4).

505        The choice of IOP algorithm to retrieve $b_{bp}(\lambda)$ is key for the PSD/phyto C algorithm, as the spectral shape of $b_{bp}$ is what the
PSD slope retrieval is based upon (Eq. 4). The Loisel and Stramski (2000) IOP algorithm is chosen here, as in KSM09, because
it allows spectral $b_{bp}$ retrievals that are not constrained by a specific spectral function or parameterization on $b_{bp}$ as is done,
for example, in QAA (Lee et al., 2002) and GSM (Maritorena et al., 2002, 2010). For the wavelengths used in the PSD slope
retrieval, modeled and satellite-derived $b_{bp}$ spectral shapes compare well when the Loisel and Stramski (2000) algorithm is
used and global patterns of the retrieved PSD parameters appear reasonable. Preliminary tests with Loisel et al. (2018) indicate
that this algorithm is not as suitable for PSD retrieval in this regard. Use of Loisel et al. (2018), Jorge et al. (2021) and other
IOP algorithms will be further investigated in future development of the PSD algorithm.

        An important assumption of the model is that $N_0$ for NAP is twice that for phytoplankton, so that the phyto C to POC ratio
is a constant 1:3. This ratio is expected to vary in the real ocean, and the value used here is a reasonable average choice (e.g.
Behrenfeld et al. (2005); Jackson et al. (2017); Thomalla et al. (2017) and refs. therein). Graff et al. (2012) employed the cell
sorting and chemical analysis methods of Graff et al. (2012) to measure phyto C in the Equatorial Pacific and along the Atlantic
Meridional Transect (AMT). Their results indicate that a phyto C:POC value of 1/3 is reasonable, falling within their observed
ranges; however, they do observe many higher values, particularly in the oligotrophic gyres. The Roy et al. (2017) phyto C to
Stramski et al. (2008) POC ratio (applied to the May 2015 monthly image of the OC-CCI v5.0 data) indicates generally lower
values of this ratio (with some high latitude and coastal exceptions), and even lower values occur in the gyres, with values
mostly below 0.1 in the low-latitude open ocean (not shown). In light of this observation, note the difference between the Graff
et al. (2015) and Roy et al. (2017) phyto C retrievals (Fig. 9 and Supplement Fig. S7). Further direct analytical observations of
phyto C and the reconciliation and better understanding of the spatio-temporal variability of the phyto C to POC ratio should
be a high priority in order to improve understanding of carbon pools and their relationships in the ocean (Brewin et al., 2021)
and to retrieve phyto C reliably from space.

        When the modeled $b_{bp}$ spectra (Fig. 2) are used to design look-up tables (LUTs) as in KSM09 (Kostadinov et al., 2009), one
can compare the KSM09 algorithm to the model developed here. Results indicate that the LUTs relating the $b_{bp}$ slope to the





PSD slope are quite similar, differing by $\approx 0.2$ (in PSD slope) at the most, when $\xi$ values are from 3.5 to 4.0 (not shown). The LUTs practically coincide for very low and very high PSD slopes. LUTs of the new algorithm for the various sensors are very

similar to each other. The LUTs linking the $b_{bp}$ slope to the $N_0$ parameter for KSM09 and the new algorithm are also similar to first order (in logarithmic space). Importantly, the new algorithm LUTs (also nearly coinciding for the various sensors) indicate higher backscattering per particle for low $b_{bp}$ slope values below $\approx 0.75$ (typical for more eutrophic waters), and they indicate lower backscattering per particle for steeper backscattering spectral slopes (typical for more oligotrophic conditions). This can lead to up to a factor of $\approx 2$ difference in retrieved $N_0$, i.e. in particle concentrations retrieved. This LUT difference leads to a

reduction of the apparent range exaggeration of retrieved phyto C globally (low values in the subtropical gyres, and high values in the eutrophic areas). It is this range exaggeration that led to the need for the empirical tuning in Kostadinov et al. (2016a). While the need for this tuning seems to persist for the new algorithm and is also implemented here (Sec. 3.3), validation and comparisons results here suggest that for some variables and algorithms being compared with, the original version of the novel PSD/phyto C algorithm performs better.

The pico-phytoplankton C data in Martínez-Vicente et al. (2017) are derived from cell counts (abundance) converted to carbon using specific conversion factors for different species/groups. Namely, 60 fg C per cell was used for *Prochlorococcus*, 154 fg cell$^{-1}$ - for *Synechococcus*, and 1319 fg cell$^{-1}$ for pico-eukaryotes. This differs from the PSD-based phyto C retrieval algorithm in which the conversion is a function of cell volume and is continuous. For the allometric coefficients of Roy et al. (2017) used here, the equivalent conversion factor is $\approx 53$ fg cell$^{-1}$ for cells of the smallest diameter within the pico-

phytoplankton range (0.5 μm), and is $\approx 1825$ fg cell$^{-1}$ for the largest diameter cells within the pico-phytoplankton range (2.0 μm), indicating first order consistency, but not full equivalency, with the methods of Martínez-Vicente et al. (2017).

   The power law (Eq. 1) is a parameterization of real-world PSDs, and while there are theoretical underpinnings (e.g. West et al. (1997); Brown et al. (2004); Hatton et al. (2021)) and observations (e.g. Quinones et al. (2003)) supporting its applicability over large size ranges, real-world PSDs may deviate from the power-law, especially in coastal zones (e.g. Reynolds et al.

(2010); Buonassissi and Dierssen (2010); Koestner et al. (2020)). The power-law is not a converging PSD model, i.e. it is sensitive to the chosen limits of integration (for a sensitivity analysis to the integration limits, see Kostadinov et al. (2016a)). Gamma functions may be a better choice to represent marine PSDs (Risović, 1993; Risović, 2002). However, we choose to use the power-law because of its theoretical underpinnings and because the goal is to build an operational algorithm for existing multispectral data with limited degrees of freedom. We additionally assume that the PSD slope for both phytoplankton and

NAP is the same, limiting the number of parameters to be retrieved. Hyperspectral data and observations of phytoplankton and NAP-specific PSDs will be needed to relax these assumptions in the future. Organelli et al. (2020) observed that the PSD slope steepened for small particles, deviating from a power-law. This steepening could partially explain the putative under-estimates of the original algorithm in oligotrophic gyres. Moreover, the absolute number of particles retrieved is sensitive to uncertainties in the real index of refraction assumed. While the goal here is to create a global algorithm which uses one set of end-members,

we recognize that future implementations can be improved by assessing the impact of using regionally variable subsets of index of refraction distributions. The PSD parameterization and choices of Mie inputs, in particular complex indices of refraction, represent important sources of uncertainty and can also affect the need for tuning and the degree of suitability of estimating





POC with our generic NAP population. Further algorithm analysis of performance and improvements need to focus on the index of refraction choices for the particle populations. For further discussion of algorithm uncertainties, see Kostadinov et al.

(2009),Kostadinov et al. (2010) and Kostadinov et al. (2016a).

Graff et al. (2015) observe a relationship between phyto C and $b_{bp}$ that is stronger than that for other proxies. This is encouraging for the use of backscattering as a proxy for phytoplankton carbon biomass. However, the link between the PSD and $b_{bp}$ spectral shape is a second-order effect that is not easily observed in *in-situ* observations (Kostadinov et al., 2009; Slade and Boss, 2015; Organelli et al., 2020), even though theoretical modeling demonstrates a clear link (Kostadinov et al. (2009);

this study). Kostadinov et al. (2012) discuss some reasons why it may be difficult to observe this relationship in current *in-situ* data, e.g. the fact that the PSD is fit over a narrow range of diameters compared to the size range theoretically affecting $b_{bp}$. Nevertheless, these considerations and the overall performance of the KSM09 homogeneous algorithm as compared to the algorithm presented here leads to the conclusion that there are four primary directions that should be priorities for moving forward. First, investigate the effect of choices of index of refraction distributions, as discussed above. Second, rather than

relying only on $b_{bp}$ for PSD and phyto C retrieval, a blended approach should be developed that also uses absorption, i.e. combine the approach here with that of Roy et al. (2017). Third, investigate the ability of hyperspectral data to provide more degrees of freedom for retrieval of more variables simultaneously, allowing relaxation of some key assumptions and perhaps a third particle population to represent POC and mineral particles separately; this is important in light of the upcoming PACE mission (Werdell et al., 2019). Hyperspectral absorption data in particular have the potential to increase information content

and allow group-specific retrievals (e.g. Kramer et al. (2022), but see also Cael et al. (2020)). Finally, fourth, collect more global, comprehensive *in-situ* data sets of all relevant variables, including and especially of phyto C (Graff et al. (2015)), for further model development and validation. With regard to the latter, agencies and investigators should focus on building quality controlled, one-stop-shop data sets.

## 4 Additional Information

*Code and data availability*. Code and data associated with algorithm development, as well as operational application to OC-CCI v5.0 data are published on the Zenodo® repository (Kostadinov et al., 2022) and are available at the following URL: https://doi.org/10.5281/zenodo.6354654.

An OC-CCI v5.0-based satellite PSD/phyto C data set (monthly, 1997-2020, plus monthly and overall climatologies) has been published on the PANGAEA® repository (Kostadinov et al., 2022) and is freely available in netCDF format and browse images at the following URL:

https://doi.org/10.1594/PANGAEA.939863



**Appendix A: Details on the OC-CCI v5.0 Dataset**

Processing and analysis was done using the sinusoidal projection of OC-CCI v5.0. For user convenience, once the final products were generated, they were re-projected to equidistant cylindrical projection (unprojected latitude/longitude) before publication to the data repository linked above (Sec. 4). The empirical tuning (Sec. 3.3) is not applied to the variables in the published

data set (Sec. 4). Instead, the spatially-explicit linear-space multiplicative tuning factor (Supplement Fig. S6B) is given. The choice to provide an optional tuning to be applied at the user's discretion is dictated by the validation and comparison results discussed in the manuscript.

*Author contributions.* TSK designed the study, conducted the modeling and algorithm development and data analyses, and wrote the manuscript. SB and LRL provided the EAP model code and technical support for EAP modeling. SM helped with error propagation es-

timates and designed the band shifting methodology. CEK, BJ, VMV and SS extracted match ups and/or provided match up *in-situ* and satellite data set compilations. XZ provided the coated spheres code and technical support for it, as well as validation PSD data. HL and DSFJ provided technical assistance with IOP code testing. EK tested backscattering spectral shape sensitivity. SR provided Roy et al. (2017) algorithm output data. SB, LRL, XZ, SM, EK, SR, CEK, BJ, SS, HL, and DSFJ read the manuscript and provided comments/edits.

*Competing interests.* The authors declare no competing interests.

*Disclaimer.* The views and opinions expressed here are those of the authors and do not necessarily express those of NASA.

*Acknowledgements.* Funding for this project was provided by NASA grant #80NSSC19K0297 to TSK, Irina Marinov and SM. TSK also acknowledges support from California State University San Marcos. This work is a contribution to the Simons Foundation Project Computational Biogeochemical Modeling of Marine Ecosystems (CBIOMES, number 549947, S.S.). This paper is also a contribution to the ESA projects Ocean Colour Climate Change Initiative (OC-CCI) and the Biological Pump and Carbon Exchange Processes (BICEP). Additional

support from the National Centre for Earth Observations (UK) is also gratefully acknowledged.

We acknowledge Olaf Hansen, Harish Vedantham, Marco Bellacicco, Salvatore Marullo, Irina Marinov, Ivona Cetinic, Giorgio Dall'Olmo and David Desailly for various help/useful discussions. We acknowledge the ESA OC-CCI and BICEP Project teams and contributors, *in-situ* PSD validation data contributors as given in Kostadinov et al. (2009) and David Siegel/ UCSB ERI / Plumes and Blooms Project team, the NASA EXPORTS teams, and Giorgio Dall'Olmo, Emmanuele Organelli and the AMT26 cruise team for their in-situ PSD data (Organelli and

Dall'Olmo, 2018). We acknowledge all *in-situ* data contributors to the BICEP/POCO projects compilations of POC and pico-phytoplankton carbon data sets. The OC-CCI reference is Sathyendranath et al. (2019) and the v5.0 specific reference is Sathyendranath et al. (2021). The modeling and processing is done using the sinusoidal projection (one of the projections provided by OC-CCI), whereas maps here are presented in equidistant cylindrical projection (unprojected lat/lon). Erik Fields, ESA, BEAM (Brockmann Consult GmbH) and NASA are





acknowledged for the re-projection algorithm. Coastlines in maps shown here are from v2.3.7 of the GSHHS data set - see Wessel and Smith
(1996). The NOAA ETOPO1 data set (https://www.ngdc.noaa.gov/mgg/global/ was used in validation for bathymetry masking. Modeling
and data processing was done in MATLAB®.



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
