# Peer review of "Ocean Color Algorithm for the Retrieval of the Particle Size Distribution and Carbon-Based Phytoplankton Size Classes Using a Two-Component Coated-Spheres Backscattering Model"

_EGUsphere, 2022_

## Referee Comment (RC2)

Review of 'Ocean color algorithm for the retrieval of the particle size distribution and carbon-based phytoplankton size classes using a two-component coated-spheres backscattering model'.
Reviewer: Emmanuel Boss, UMaine

This paper report on the design of a new model to invert remote-sensing inversion to size. Result suggest it is not ready to be implemented in its current form.

The paper is well written. It is of significant interest. It does represents a very significant effort.

However, my biggest fear is, as happened with the previous versions of this model, that it will be implemented by modelers of ocean BGC to make predictions on ecosystems, export etc' while not propagating the large biases observed in the validation of this paper. I do realize it is not my job to protect the community from poor use of biased models.

For it to be more useful it needs, in my mind, additional work.
1. There is significantly more data for validation than suggested here. For example, there is LISST data from NAAMES and EXPORTS from my group on Seabass as well as direct observations of phytoplankton PSDs (reported in Haentjens et al., 2022).
2. As Organelli et al., 2018 have shown, the same parameters retrieved here could be used to predict the beam attenuation measured by, for example, the LISST, C-star and AC-S instruments. We have many more data of all those from the whole ocean (e.g. Tara datasets on SeaBASS). Why not use them as part of your validation? It could help you to better constrain model parameters (particularly associated with NAP).
3. Chlorophyll itself is a predictor of PFT, PCT and PSD. I would like to see, as I discussed with the lead author in the past, proof that the prediction of this model are significantly different than relationship with Chlorophyll itself or other current product from Rrs (e.g. bbp). Why go for a complicated model if a simple one is just as good (or just as bad, depending on your point of view)? I would love to see property-property plots involving C_phyto/POC, Chl, No and \xsi. In short, does the inclusion of all the model parameters into a novel model able to teach us about the ocean in ways [chl] does not (all the global maps presented seem highly correlated with [chl] and/or bbp distribution in the upper ocean)?
4. It is obvious that No and \xsi are correlated. Why not show their relationship? Is it consistent with in-situ data?
5. The use of the same satellite matchup for multiple validation seems not acceptable to me as it is not an independent evaluation. You could/should average the in-situ data prior to matching up.
Minor comments.
1. Figures numbers are not consistent with order of their citation.
2. Line 1445: remove 'using'.

Using the rubric of EGU:
Scientific significance -3

Scientific quality -3
Presentation quality – 2

1. Does the paper address relevant scientific questions within the scope of OS?

   yes

2. Does the paper present novel concepts, ideas, tools, or data?

   Yes.

3. Are substantial conclusions reached?

   Not really, in my mind, beyond those of model evaluation.

4. Are the scientific methods and assumptions valid and clearly outlined?

   Clearly outlined, but validity needs further work.

5. Are the results sufficient to support the interpretations and conclusions?

   No.

6. Is the description of experiments and calculations sufficiently complete and precise to allow their reproduction by fellow scientists (traceability of results)?

   Yes.

7. Do the authors give proper credit to related work and clearly indicate their own new/original contribution?

   Yes.

8. Does the title clearly reflect the contents of the paper?

   Yes.

9. Does the abstract provide a concise and complete summary?

   Yes.

10. Is the overall presentation well structured and clear?

Yes.

11. Is the language fluent and precise?

    Yes.

12. Are mathematical formulae, symbols, abbreviations, and units correctly defined and used?

    Yes.

13. Should any parts of the paper (text, formulae, figures, tables) be clarified, reduced, combined, or eliminated?

    See above comments.

14. Are the number and quality of references appropriate?

    Yes.

15. Is the amount and quality of supplementary material appropriate?

    Yes.

---

## Author Comment (AC1)

**Reviewer # 1**

**General comment**

This study attempts to retrieve from space (ocean color satellite data) information on particle size distribution and carbon-based phytoplankton size classes in open ocean waters. This significant piece of work is actually the extension of previous studies (Kostadinov et al. 2007-2022) which includes validation results. The manuscript is well organized, written and illustrated.

Thank you.

Unfortunately, these validation results are not convincing, most probably as several assumptions made in the methodology are not valid. The authors should carefully revise the assumptions made notably to model the particle size distributions and discuss the impact on the resulting satellite-derived products. Detailed comments are provided hereafter to clarify the methodology and discuss the validation results.

While we agree that the regression statistics of the PSD validation exhibit low $R^2$ values, they are statistically significant, and we posit that there can be multiple reasons for that that need to be carefully considered. For example, mismatch in spatial and temporal scales of sampling between satellite vs. in-situ data is a possible reason for poor validation. More reasons are discussed below, and also are covered in more detail in Kostadinov et al. (2009).

[Figure]

[Figure]

Note that the $\xi$ vs. $N_0$ relationship in in-situ data (only those that are matched up are shown) is rather weak and driven only by a few outliers. When these are removed, there is no statistically significant linear relationship between the two PSD parameters in the in-situ data (slope 95% confidence interval crosses zero, $R^2 << 0.01$) (See the regression plot above with outliers excluded; interestingly, most of these outliers come from one area – the California Bight). Thus, while the satellite data, as expected, exhibit strong negative correlation between the PSD slope and $N_0$, the in-situ data exhibit a weak positive correlation that disappears with removal of a few outliers. See above the bivariate histogram of satellite data (OC-CCI v.5.0 climatology using the new algorithm presented here), with in-situ match-ups superimposed. While one possible reason for that can indeed be that the satellite algorithm uses wrong/incorrect model/parameterizations, we posit that there are multiple additional or alternative reasons for this discrepancy. One such important reason is the large difference of spatio-temporal coverage between the satellite and the in-situ data sets. Geographic and temporal coverage of the in-situ data is rather limited, especially when matched up with satellite data (only 167 PSD match-ups are available after the match-ups were re-done per request of the other reviewer, some more are expected to be added in revised manuscript, see comments to other reviewer). This suggests that the in-situ data is not necessarily able to capture a relationship that the satellite data does. We note that this relationship has a theoretical underpinning because of at least two reasons: a) the power-law tends to apply over large size ranges (e.g. Hatton et al., 2021), for which there are theoretical reasons (e.g. Brown et al, 2004) , and b) what we know about global ocean ecosystems, namely that oligotrophic areas exhibit relative dominance of smaller phytoplankton, as opposed to increased importance of larger phytoplankton in more eutrophic areas. Because of b) above, we expect backscattering in the ocean to become "bluer", i.e. to have a steeper spectral slope, in oligotrophic areas. This is indeed observed in satellite data and is qualitatively interpreted in this way in Loisel et al. (2006). Therefore, we expect, in the ocean, globally and on average, $N_0$ to decrease with increasing $\xi$. This is not necessarily going to be captured by in-situ data of limited spatio-temporal coverage that is also fit over a relatively small size range as compared to the full optically significant size range. This latter point brings us to another possible reason why in-situ data (which has its own limitations and uncertainties) may not correspond well to satellite-derived PSDs.

We note that the in-situ data used in the validation, especially for PSD, has its own limitations and should not unequivocally be considered the "sea-truth" without any reservations. In the case of PSD It is assembled from two different instruments with quite different principles of operation, and the power-law fit is done only in the range of 2 to 20 micrometers diameter, much smaller than the optically significant range. As noted by the reviewer, the power-law fit is not always great, and it can be sensitive to the diameter range used to fit it. The relationship of satellite PSD slope retrieved to the PSD slopes measured in-situ is therefore expected to be noisy, as observed. In fact, other investigations have also not always been able to demonstrate a strong or any relationship between PSD and bbp shape – e.g. Kostadinov et al., 2012; Organelli, 2020, but see also Slade and Boss, 2015. The overall trend/tendency of higher or slower slope is, however, captured, in our validation, which in our view renders the PSD slope validation reasonable, but with improvements desirable in future implementations.

In short, we believe that the relatively poor validation result and the above satellite to in-situ correlation discrepancy does not necessarily mean the satellite algorithm is "wrong". We fully acknowledge that the power-law PSD has its limitations, and more of the assumptions/limitations of the algorithm are ideally going to be relaxed in future work. Global relationships averaged over a month or longer vs. a measurement in single point in time and space are likely to work better.

Regarding the validation with POC and pico-phytoplankton C data, and the tuning procedure, it is of importance that similar tuning procedure was also implemented in an earlier version of the algorithm, Kostadinov et al. (2016). This earlier version used only one particle population and modeled it as homogeneous spheres. The algorithm construction here is substantially improved since it uses two different particle populations and a more realistic representations of phytoplankton cells. However, in much of the open ocean (especially the oligotrophic areas), the backscattering optical signal is still dominated by the NAP represented by homogeneous spheres and spanning a larger size range than phytoplankton. One and the same Monte Carlo range of Mie inputs (e.g. real indices of refraction) may not be realistic globally, from subtropical oligotrophic to coastal, to Southern Ocean vs. Northern Hemisphere high latitudes, due to , for example, non-uniform dust input in the ocean, and proximity to coasts. This likely needed regionalization/improvement is out of scope here, but should be investigated in the future.

Some of the assumptions made are necessitated by the goal to have an operational algorithm able to be applied to current multi-spectral satellite data. It is also a goal to have an algorithm based on first principles as much as possible (as mechanistic as possible). This represents a compromise. The goal to have an operational algorithm limits the degrees of freedom and number of independent variables possible to retrieve. More discussion is provided in response to your specific comments below. We will also add text in the manuscript emphasizing some of the key points of our responses here to make model limitations clearer to the user.

**Detailed comments**

Line 45, Equation 1:

To my knowledge this very convenient power law size distribution of particles does not apply to phytoplankton particles in oceanic waters. Can you please provide relevant references to support your statement?

Line 53:

Again, probably the main/major issue in this study: phytoplankton cells in oceanic waters DO NOT follow a power-law PSD. If I am wrong please prove it based of already published quality field data.

We respond to the previous two comments together as they address the same issue.

While the reviewer is correct that the PSD of a specific species is not likely to follow a power-law and will be expected to instead peak in the characteristic size of that species, here we are aiming to model the global ocean ecosystem as a whole, over large spatio-temporal scales and on average, as well as for all species present. Individual cases, especially during mono-specific strong blooms, are likely to deviate from the power law (e.g. Reynolds et al., 2010 for coastal waters), but ecosystems as a whole, especially on average and globally, are more likely to follow the power law more closely (e.g. Hatton et al., 2021).

There has been a lot of theoretical work on the power-law, particularly as applied to size of organisms and ecosystems, and the power-law has theoretical foundations/underpinnings (e.g. Brown et al., 2004; Hatton et al., 2021 and refs. therein). There's also been investigations demonstrating the power-law does not apply well, especially in coastal ecosystems (e.g. Reynolds et al., 2010; Runyan et al., 2020; Reynolds et al., 2021 and refs. therein). It is a lot harder to find information and data on living phytoplankton only, and their specific PSDs, because it has been historically difficult to separate living phytoplankton and measure, say, their PSD or carbon (e.g. Graff et al. 2012, 2015). This makes validation of one of our main products – phytoplankton C, difficult. This is mentioned in the paper.

A recent study (Haentjens et al., 2022) investigated in-situ measurements of phytoplankton-specific PSDs. They do fit their data to a power-law, but more importantly, their figures illustrate that to first order, the phytoplankton-specific PSD shape is consistent with a power-law. For example, see their Fig. 3, Fig. 5, and Supplement Fig. S8. They do state that the drop off for the smallest size bin is real due to lack of *Prochlorococcus*, but could also be an instrument/methodology artifact. We acknowledge that a drop-off in the size distribution will be expected at the limits of the size range of autotrophs, hence the power-law is not expected to apply equally well over the same size range everywhere and always in the global ocean. Again, our algorithm aims to capture first order effects and be applicable globally; regionalization may be needed to address this further, e.g. build an algorithm with different modeled size limits for phytoplankton in different regions.

Hatton et al. (2021) offer an assessment of the PSD of marine life over a huge range of sizes (body mass), demonstrating that a specific power law applies, in the context of the Sheldon (1972) hypothesis that equal biomass tends to occur in each logarithmically-spaced size bin. This also follows from our Eq. 1, for a specific PSD slope ($\xi = 4$), and when the upper and lower limits of the size bins follow a geometric progression. We note that the assessment of Hatton et al (2021) offers a strong support for the power law (their Fig. 1), and also separately analyzes autotrophs from other marine organisms with overlapping size ranges. Note from their Fig. 2a that to first order, approximately equal autotroph biomass occurs in logarithmically spaced size bins, consistent with a

power-law for phytoplankton alone and consistent with Sheldon's hypothesis (for the ocean as a whole). We emphasize here "to first order" – over larger size ranges and on average for global oceanic ecosystems. We acknowledge that individual cases in space time may not closely conform to a power-law, hence giving one possible explanation (out of several) for the relatively poor validation.

Quinones et al. (2003) investigate the size spectra of planktonic biomass and find that the power-law is followed at all stations, and biovolume-wise the Sheldon hypothesis is followed in terms of slope. We also note that Fig. 2 of Lombard et al. (2019) demonstrates that to first order the power law applies to large size ranges of planktonic organisms, sampled with different instruments. We acknowledge that phytoplankton share their size domain with other organisms (bacteria on the low end and zooplankton at the high end). So, it may be reasonable to accept that the contribution of phytoplankton to the size classes would fall off at the two ends of their range – but see the Hatton et al. (2021) discussion above.

While we acknowledge the limitations of the power-law (e.g. Reynolds et al., 2021 and refs therein; Bernard et al., 2007; Reynolds et al., 2010; Organelli et al., 2020 – see slope for small particle sizes), in spite of support for it from the literature as shown above (see also, for example, Buonassissi and Dierssen 2010), we would also like to present here a strong motivation for using the power-law PSD, as well for using the same PSD slope for both phytoplankton and NAP (which assumption is also questioned by the reviewer later on). A primary goal of this work is to build a model that is based on first principles as much as possible, and of course to stay as close to reality as possible. Some of the assumptions are hard to verify due to lack of global data sets of the variables involved. Importantly, a second major goal of the manuscript is to develop an operational algorithm that is possible to apply to current mainstream, multispectral global ocean color data, e.g. the OC-CCI data set. That puts a limit to the degrees of freedom available for retrievals, especially since only certain wavelength ranges work well with the satellite bbp retrievals we use (the choice of wavelengths is important – see for example Organelli et al. 2020). Ideally, of course, it would be good to model phytoplankton, NAP, and perhaps a third particle population to represent mineral contributions, fully independently and with more complex PSDs as needed, however, these parameters would not be possible to retrieve realistically from multispectral data.

In light of the above discussion, we note that our algorithm ultimately retrieves one slope and $N_0$ value, from the total particulate backscattering. The algorithm's assumptions are then used to assign these to phytoplankton vs. NAP PSDs. We agree that these are restrictive and not fully tested assumptions. However, adding two power-law PSDs with different slopes results in a non-power law PSD and requires retrieval of more parameters, making the inversion more complex, where not only degrees of freedom but also uniqueness of solutions may become an issue and needs further investigation. Two power-law PSDs with the same slope and different N0 add to a power-law PSDs, making the problem tractable.

We are also not convinced that using a different distribution from a power law for phytoplankton is going to make a big difference at low oceanic chlorophyll concentrations.

In fact, submicron particles are important for the shape of bbp, and their treatment may turn out o be more important that the exact shape of phytoplankton PSD. For example, our modeled signal is dominated by bbp due to NAP in oligotrophic areas (see also Bellacicco et al., 2018). This covers a

large surface area of the oceans. This of course requires quantitative verification, and testing different PSD parameterizations is planned as a priority next step. The proper form of phytoplankton PSD is a difficult concern to respond to – because it is not clear what distribution phytoplankton do have in the open ocean, data on this are limited – however, see our Haentjens et al. (2022) discussion above.

We would also like to note that the phyto C: POC = 1:3 everywhere in the ocean assumptions is more of a concern and more restrictive to us than the power-law PSD form. We believe that it's a priority to address that assumptions as best as feasible in later iterations.

We will add language to the manuscript's discussion to summarize all of the above considerations.

Line 64:

"a single population of particles (approximated by homogeneous spheres)"

This is another strong assumption which definitely does not apply to phytoplankton cells in in marine waters. Please discuss it and say what is the impact in your methodology.

We note that this sentence pertains to the earlier KSM09 (Kostadinov et al., 2009) algorithm. One of the major purposes of the effort of this manuscript is exactly to relax this assumption to the extent feasible within this effort. I.e., we are no longer assuming a single population of particles like KSM09 does – here we model two separate particle populations, phytoplankton and NAP, the optical properties of which are modeled differently and separately. Importantly, phytoplankton are *not* modeled as homogeneous spheres, rather as coated spheres, to better approximate their internal structure and hence backscattering.

Line 85:

Where do minerogenic particles come from in open ocean waters?

Minerogenic particles are delivered globally via dust deposition as part of the atmospheric dust cycle. In fact, aeolian iron sources can be important biologically both in the ocean and on land - see for example, Nogueira et al. (2021), Gao et al. (2001), Mahowald et al. (2005) and Wagener et al. (2008).

We thank you for this question, it illustrates another area of possible improvement that can prove to be important – at the moment minerogenic contributions to the particle assemblage are considered a source of random uncertainly via the choice of Mie inputs and the Monte Carlo simulations, and in future iterations they could be quantified in a spatially and/or temporally explicit manner, which should improve algorithm performance and reduce uncertainty.

Line 93 'an initial effort of validation':

Such an effort to at least first validate the assumptions made in your recent and present studies and notably validate the PSD algorithm should have been made already, before going forward applying non-validated algorithms to satellite data and interpret the results obtained

We note that this is the initial paper describing the novel PSD algorithm, and it does come with a validation, including the PSD parameters and derived variables. The assets (published scientific code and data) are part of this paper. We state that it is an initial validation, because it's the first for this algorithm, and because more is planned. Further validation (including with more data and more variables) is planned, and validation results are and will be taken into account for further algorithm development. Prior iterations of the algorithm and publications (e.g. Kostadinov et al., 2009, 2010, 2016, 2017) do contain validation and/or inter-comparisons.

Line 98 'backscattering are modeled using Mie theory (Mie, 1908) for homogeneous spherical particles and the Aden-Kerker (Aden and Kerker, 1951) method for coated spheres.':

Is Mie theory well adapted to your study?

What not considering also the more realistic case of non-spherical particles?

Mie theory is applied to NAP, which are modeled as homogeneous spheres. Phytoplankton are modeled as coated spheres. The reviewer is of course right that not all particles closely conform to these spherical models, and random shoes should be considered. Considering random shapes complicates the optical solutions greatly, and it's non-trivial to find solutions that work for the needed size ranges for optical oceanography (e.g. Clavano et al., 2007). The EAP framework using coated spheres has the advantage in that it provides a realistic model for a phytoplankton cells – backscattering is significantly affected by use of coated vs. homogeneous spheres, as we show here and as is shown, for example, in Organelli et al. (2018). We agree that random-shaped particles should be considered for both NAP and phytoplankton and spherical vs. random shaped will affect backscattering (e.g. Clavano et al., 2007), but given the limitations above and the need to specify more parameters such as shapes, we consider this to be out of scope for this work, and something to be considered for future tests and iterations.

Line 138 'The two key assumptions are: 1) Phytoplankton and NAP have a power-law PSD (Eq. 1) with the same slope ξ'

Once again, I do not agree for phytoplankton. Moreover why the same slope?

We address this concern in our comment above for lines 45 and 53, where we discuss the applicability of the power-law PSD to phytoplankton.

Tables 1 and 2:

Please justify the choice of the minimum, mean and maximum values considered here as inputs. Are your computations realistic??

For many of these inputs, the main references are Robertson-Lain et al. (2018) and Bernard et al. (2009), and refs. therein, as well as Kostadinov et al. (2009) and refs. therein. We discuss additional references below. For the Dmin of phytoplankton – this is dictated by the size of the smallest autotrophs, *Prochlorococcus*, which tend to have diameter of about 0.5 μm or slightly larger (e.g. Morel et al., 1993). For the largest autotrophs it would depend on ecosystem and is harder to pick, hence

we use a distribution, but as a guidance, individual cells larger than ~ 50 μm in diameter are rare/not expected to be found in the open ocean (e.g. Charles Stock, pers. comm to Kostadinov et al., 2016 author team). Our algorithm is meant for global open ocean applications, not to a particular ecosystem, thus the mean Dmax is meant to be close to that value.

For the Dmin and Dmax of NAP – see Duforet-Gaurier et al. (2018), their Table 1. For NAP, since it's important to capture optically active range of particles, this was investigated with the bbp cumulative plots – our Fig. S4. For this reason, we use Dmin = 0.01 mm for NAP, unlike Duforet-Gaurier et al. (2018). If sub-micron-sized NAP is not included, the steeper end of bbp shapes observed in satellite data cannot be reproduced by the model. Note that Stramski and Kiefer (1991) use an even lower limit of 0.002 μm.

For the indices of refraction – apart from Robertson-Lain et al. (2018) and Bernard et al. (2009), see also Morel and Bricaud 1986, Babin et al. (2003), Wozniak and Stramski (2004), and Duforet-Gaurier et al. (2018).

We will add more references to the table to clarify sources.

Line 218:

Define LISST

The acronym will be defined in the revised manuscript, thank you!

Figure 8 'PSD validation results':

Thank you for showing these validation results which are not satisfactory, as could be expected considering that several assumptions made are (most probably) not valid.

While there is somehow an agreement (or at least a trend) between the satellite and situ No (number of particles), there is no correlation for the slope, therefore no validation of the satellite-derived PSD, assuming the PSD is a power-law.

We note that the PSD slope validation regression is statistically significant, albeit with a low $R^2$ value. As for N0, we note that the algorithm is able to pick up the concentration of particles, to first order, according to this validation. We find this to be impressive and consider it a success, given that the algorithm makes no a-priori prescriptions about particle concentrations – they are solved for from the magnitude and shape of satellite bbp. The algorithm also allows for a wide variation of real indices of refraction, which results in large uncertainties in N0. We further comment on the PSD validation result in response to your general comment at the beginning of this review, also commenting on why in-situ data may not capture the global satellite patterns.

These poor validation results must be discussed so as its implication on the whole methodology. What would be the results if another (more realistic) function was used to model the PSD?

We will add relevant discussion about the PSD parameterization and as stated elsewhere in our responses.

Figure 10:

These validation results are more convincing. Please specify in the figure legend what you mean by 'empirical tuning'.

Thank you, we will edit the figure legend to include explicit description of the empirical tuning for $N_0$.

Figure 11.

As in Figure 8, poor validation results.

For this validation, as with the PSD data, the in-situ data set quality, method of derivation, and limitations, as well as spatio-temporal coverage have to be considered. We note, for example, that these are not direct phytoplankton carbon measurements. Albeit poor (and not significant in the case of the tuned version), these results are extremely important to report, as, unlike the rest of the phytoplankton C (and Chl) validations and verifications we present, these results are not improved by the tuning, making it ambiguous whether the tuning should be applied. We believe that the continued need for tuning is a primary issue to resolve foe future improvements, and by presenting all these results together, we aim to prompt the community to work in these directions too. Addressing the tuning further may require not only considering different PSD shapes as you suggest, but also, importantly, considering the distributions of the Mie inputs and that they may need regionalization. The latter, we suspect, is more important.

See also relevant parts of the PSD validation responses we offer above.

References:

Babin, M., Morel, A., Fournier-Sicre, V., Fell, F., & Stramski, D. (2003). Light scattering properties of marine particles in coastal and open ocean waters as related to the particle mass concentration. *Limnology and Oceanography*, *48*(2), 843–859. https://doi.org/10.4319/lo.2003.48.2.0843

Bellacicco, M., Volpe, G., Briggs, N., Brando, V., Pitarch, J., Landolfi, A., Colella, S., Marullo, S., & Santoleri, R. (2018). Global Distribution of Non-algal Particles From Ocean Color Data and Implications for Phytoplankton Biomass Detection. *Geophysical Research Letters*, *45*(15), 7672–7682. https://doi.org/10.1029/2018GL078185

Bernard, S., Shillington, F. A., & Probyn, T. A. (2007). The use of equivalent size distributions of natural phytoplankton assemblages for optical modeling. *Optics Express*, *15*(5), 1995. https://doi.org/10.1364/oe.15.001995

Brown, J. H., Gillooly, J. F., Allen, A. P., Savage, V. M., & West, G. B. (2004). Toward a metabolic theory of ecology. In *Ecology* (Vol. 85, Issue 7).

Buonassissi, C. J., & Dierssen, H. M. (2010). A regional comparison of particle size distributions and the power law approximation in oceanic and estuarine surface waters. *Journal of Geophysical Research: Oceans*, *115*(10), 1–12. https://doi.org/10.1029/2010JC006256

Clavano, W., Boss, E., & Karp-Boss, L. (2007). *Inherent Optical Properties of Non-Spherical Marine-Like Particles ‚Äî From Theory To Observation*. 1–38. https://doi.org/10.1201/9781420050943.ch1

Duforêt-Gaurier, L., Dessailly, D., Moutier, W., & Loisel, H. (2018). Assessing the impact of a two-layered spherical geometry of phytoplankton cells on the bulk backscattering ratio of marine particulate matter. *Applied Sciences (Switzerland)*, *8*(12). https://doi.org/10.3390/app8122689

Gao, Y., Kaufman, Y. J., Tanré, D., Kolber, D., & Falkowski, P. G. (2001). Seasonal distributions of Aeolian iron fluxes to the global ocean. *Geophysical Research Letters*, *28*(1), 29–32. https://doi.org/10.1029/2000GL011926

Graff, J. R., Milligan, A. J., & Behrenfeld, M. J. (2012). The measurement of phytoplankton biomass using flowcytometric sorting and elemental analysis of carbon. *Limnology and Oceanography: Methods*, *10*(NOVEMBER), 910–920. https://doi.org/10.4319/lom.2012.10.910

Graff, J. R., Westberry, T. K., Milligan, A. J., Brown, M. B., Dall'Olmo, G., van Dongen-Vogels, V., Reifel, K. M., & Behrenfeld, M. J. (2015). Analytical phytoplankton carbon measurements spanning diverse ecosystems. *Deep-Sea Research Part I: Oceanographic Research Papers*, *102*, 16–25. https://doi.org/10.1016/j.dsr.2015.04.006

Haëntjens, N., Boss, E. S., Graff, J. R., Chase, A. P., & Karp-Boss, L. (2022). Phytoplankton size distributions in the western North Atlantic and their seasonal variability. *Limnology and Oceanography*, *67*(8), 1865–1878. https://doi.org/10.1002/lno.12172

Hatton, I. A., Heneghan, R. F., Bar-On, Y. M., & Galbraith, E. D. (2021). The global ocean size spectrum from bacteria to whales. *Science Advances*, *7*(46), 1–13. https://doi.org/10.1126/sciadv.abh3732

Kostadinov, T. S., Siegel, D. A., & Maritorena, S. (2009). Retrieval of the particle size distribution from satellite ocean color observations. *Journal of Geophysical Research: Oceans*, *114*(9). https://doi.org/10.1029/2009JC005303

Kostadinov, T. S., Siegel, D. A., & Maritorena, S. (2010). Global variability of phytoplankton functional types from space: Assessment via the particle size distribution. *Biogeosciences*, *7*(10). https://doi.org/10.5194/bg-7-3239-2010

Kostadinov, T. S., Siegel, D. A., Maritorena, S., & Guillocheau, N. (2012). Optical assessment of particle size and composition in the Santa Barbara Channel, California. *Applied Optics*, *51*(16). https://doi.org/10.1364/AO.51.003171

Kostadinov, T. S., Milutinovic, S., Marinov, I., & Cabré, A. (2016). Carbon-based phytoplankton size classes retrieved via ocean color estimates of the particle size distribution. *Ocean Science*, *12*(2). https://doi.org/10.5194/os-12-561-2016

Kostadinov, T. S., Cabré, A., Vedantham, H., Marinov, I., Bracher, A., Brewin, R. J. W., Bricaud, A., Hirata, T., Hirawake, T., Hardman-Mountford, N. J., Mouw, C., Roy, S., & Uitz, J. (2017). Inter-comparison of phytoplankton functional type phenology metrics derived from ocean color algorithms and Earth System Models. *Remote Sensing of Environment*, *190*. https://doi.org/10.1016/j.rse.2016.11.014

Loisel, H., Nicolas, J. M., Sciandra, A., Stramski, D., & Poteau, A. (2006). Spectral dependency of optical backscattering by marine particles from satellite remote sensing of the global ocean. *Journal of Geophysical Research: Oceans*, *111*(9), 1–14. https://doi.org/10.1029/2005JC003367

Lombard, F., Boss, E., Waite, A. M., Uitz, J., Stemmann, L., Sosik, H. M., Schulz, J., Romagnan, J. B., Picheral, M., Pearlman, J., Ohman, M. D., Niehoff, B., Möller, K. O., Miloslavich, P., Lara-Lopez, A., Kudela, R. M., Lopes, R. M., Karp-Boss, L., Kiko, R., … Appeltans, W. (2019). Globally consistent quantitative observations of planktonic ecosystems. In *Frontiers in Marine Science* (Vol. 6, Issue MAR). Frontiers Media S.A. https://doi.org/10.3389/fmars.2019.00196

Mahowald, N. M., Baker, A. R., Bergametti, G., Brooks, N., Duce, R. A., Jickells, T. D., Kubilay, N., Prospero, J. M., & Tegen, I. (2005). Atmospheric global dust cycle and iron inputs to the ocean. In *Global Biogeochemical Cycles* (Vol. 19, Issue 4). https://doi.org/10.1029/2004GB002402

Morel, A., & Bricaud, A. (1986). Inherent properties of algal cells including picoplankton: theoretical and experimental results. *Canadian Bulletin of Fisheries and Aquatic Sciences*, *214*, 521–559.

Morel, A., Yu-Hwan Ahn, Partensky, F., Vaulot, D., & Claustre, H. (1993). Prochlorococcus and Synechococcus: a comparative study of their optical properties in relation to their size and pigmentation. *Journal of Marine Research*, *51*(3), 617–649. https://doi.org/10.1357/0022240933223963

Nogueira, J., Evangelista, H., Valeriano, C. de M., Sifeddine, A., Neto, C., Vaz, G., Moreira, L. S., Cordeiro, R. C., Turcq, B., Aniceto, K. C., Neto, A. B., Martins, G., Barbosa, C. G. G., Godoi, R. H. M., & Shimizu, M. H. (2021). Dust arriving in the Amazon basin over the past 7,500 years came from diverse sources. *Communications Earth and Environment*, *2*(1). https://doi.org/10.1038/s43247-020-00071-w

Organelli, E., Dall'Olmo, G., Brewin, R. J. W., Tarran, G. A., Boss, E., & Bricaud, A. (2018). The open-ocean missing backscattering is in the structural complexity of particles. *Nature Communications*, *9*(1). https://doi.org/10.1038/s41467-018-07814-6

Organelli, E., Dall'Olmo, G., Brewin, R. J. W., Nencioli, F., & Tarran, G. A. (2020). Drivers of spectral optical scattering by particles in the upper 500 m of the Atlantic Ocean. *Optics Express*, *28*(23), 34147. https://doi.org/10.1364/oe.408439

Quinones, R. A., Platt, T., & Rodríguez, J. (2003). Patterns of biomass-size spectra from oligotrophic waters of the Northwest Atlantic. *Progress in Oceanography*, *57*(3–4), 405–427. https://doi.org/10.1016/s0079-6611(03)00108-3

Reynolds, R. A., Stramski, D., Wright, V. M., & Woźniak, S. B. (2010). Measurements and characterization of particle size distributions in coastal waters. *Journal of Geophysical Research: Oceans*, *115*(8). https://doi.org/10.1029/2009JC005930

Reynolds, R. A., & Stramski, D. (2021). Variability in Oceanic Particle Size Distributions and Estimation of Size Class Contributions Using a Non-parametric Approach. *Journal of Geophysical Research: Oceans*, *126*(12). https://doi.org/10.1029/2021JC017946

Runyan, H., Reynolds, R. A., & Stramski, D. (2020). Evaluation of Particle Size Distribution Metrics to Estimate the Relative Contributions of Different Size Fractions Based on Measurements in Arctic Waters. *Journal of Geophysical Research: Oceans*, *125*(6). https://doi.org/10.1029/2020JC016218

Robertson-Lain, L., & Bernard, S. (2018). The fundamental contribution of phytoplankton spectral scattering to ocean colour: Implications for satellite detection of phytoplankton community structure. *Applied Sciences (Switzerland)*, *8*(12), 1–34. https://doi.org/10.3390/app8122681

Sheldon, R. W., Prakash, A., Sutcliffe, W. H., (1972), the size distribution of particles in the ocean, *Limnology and Oceanography*, 17, doi: 10.4319/lo.1972.17.3.0327.

Slade, W. H., & Boss, E. (2015). Spectral attenuation and backscattering as indicators of average particle size. *Applied Optics*, *54*(24), 7264. https://doi.org/10.1364/ao.54.007264

Stramski, D., & Kiefer, D. A. (1991). Light scattering by microorganisms in the open ocean. *Progress in Oceanography*, *28*(4), 343–383. https://doi.org/10.1016/0079-6611(91)90032-H

Wagener, T., Guieu, C., Losno, R., Bonnet, S., & Mahowald, N. (2008). Revisiting atmospheric dust export to the Southern Hemisphere ocean: Biogeochemical implications. *Global Biogeochemical Cycles*, *22*(2). https://doi.org/10.1029/2007GB002984

Woźniak, S. B., & Stramski, D. (2004). Modeling the optical properties of mineral particles suspended in seawater and their influence on ocean reflectance and chlorophyll estimation from remote sensing algorithms. *Appl. Opt.*, *43*(17), 3489–3503. https://doi.org/10.1364/AO.43.003489

---

## Author Comment (AC2)

**Reviewer #2**

Review of 'Ocean color algorithm for the retrieval of the particle size distribution and carbon-based phytoplankton size classes using a two-component coated-spheres backscattering model'.
Reviewer: Emmanuel Boss, UMaine

This paper report on the design of a new model to invert remote-sensing inversion to size. Result suggest it is not ready to be implemented in its current form.

While we agree that the validation regression results are less than satisfactory (which has required the tuning implemented; in the case of the PSD validation regression we offer some reasons why the validation statistics are poor, in response to your comment to investigate the $\xi$ to $N_0$ relationship), we respectfully disagree that the model and operational algorithm are "not ready to be implemented." The algorithm represents a substantial under-the-hood improvement of KSM09 (Kostadinov et al., 2009) – we now model phytoplankton cells and NAP as two separate particle populations, and phytoplankton are modeled as coated spheres to better represent their optical properties. Global patterns of PSD parameters and derived variables are meaningful in the sense that they correspond to current understanding of oceanic ecosystems. We acknowledge that this is an experimental satellite product, with relatively large uncertainties for some of the retrieved variables. Our goal is to have as mechanistic, first-principles based algorithm as feasible, while keeping it operational with current multispectral data. Naturally, also our goal is to push the boundaries of what's retrievable with such satellite data. We will add language to stress that this is an experimental satellite product, and not claiming to be a canonical, thoroughly validated product. Further validation and algorithm improvement is an ongoing and future work. Further relevant comments are provided below.

The algorithm also offers new (with respect to prior algorithm versions) and very useful ancillary retrievals – Chl from the PSD and bbp partitioned to Chl and NAP. These will be further analyzed and offer opportunities for further constraining and improving the algorithm. The Chl product is already used in the tuning. We believe that these results should be reported to the community now to move the science forward.

The paper is well written. It is of significant interest. It does represents a very significant effort.

Thank you!

However, my biggest fear is, as happened with the previous versions of this model, that it will be implemented by modelers of ocean BGC to make predictions on ecosystems, export etc' while not propagating the large biases observed in the validation of this paper. I do realize it is not my job to protect the community from poor use of biased models.

This is a very good point, and we agree – in the lead author's own experience, users of satellite data who are not experts in marine bio-optics and remote sensing tend to assume the satellite data is "perfect" and models need to be tuned to it. We emphasize that this is an experimental research product, and do not make claims that it is akin in validity and accuracy to the more established (and

much more empirical!) algorithms for canonical products such as Chl and POC. We will add language to the manuscript to explicitly emphasize this and make it clear to potential users that they need to take into account product uncertainties, and algorithm assumptions. The products do come with (partial) propagated per-pixel uncertainties, which take a large effort to produce, but we consider them essential. These uncertainties should help guide users' understanding of product status. In an analogous fashion, over-trusting in-situ data when validating satellite data should be done with caution – see our comments below regarding the PSD parameters validation.

**For it to be more useful it needs, in my mind, additional work.**

See our reply above on your "not ready to be implemented" comment, as well as additional replies to your comments below. We agree that of course the algorithm needs more additional work – that is always the case (an algorithm of this nature, at this stage of the state-of-the-art, is never, in our mind, "finished") - but this step represents major changes and improvements with respect to prior versions. We believe publication of this novel algorithm at this stage will be useful also because the results (including and importantly "negative" ones such as relatively poor comparison with in-situ PSD data, continued need for tuning) will inform the team, and importantly the community for directions for further development – e.g., the phyto C:POC = 1/3 assumption, the PSD parameterization, how to deal with submicron particles, the globally used single set of Mie inputs, etc.

We do not realistically expect validation to be "perfect" or nearly so. In fact, we believe a major qualitative improvement to the approach is needed to make substantial further progress – we list some of those at the end of the paper, e.g. using phytoplankton absorption in the retrieval (blending our approach with that of Roy et al. (2017)). You also explicitly mention one such improvement – involving optical variables that are modeled by our algorithm, and are much more available globally. We thank you for this suggestion.

**1. There is significantly more data for validation than suggested here. For example, there is LISST data from NAAMES and EXPORTS from my group on Seabass as well as direct observations of phytoplankton PSDs (reported in Haentjens et al., 2022).**

We thank you for the suggestion and will add LISST data from EXPORTS and NAAMES to the PSD validation. We note that the validation is stated to be preliminary, and we do not make claims that the PSD data collected is comprehensive and includes every possible PSD measurement make globally. As you know, there's generally a dearth of PSD data, and importantly, lack of a one-stop-shop place where all such data is easily accessible in processed, merged if needed, and QCed form. As such, acquiring, quality controlling, processing and compiling PSD data for validation represents a large effort. The intention is to conduct further validations that should help inform further algorithm development in the future. This, importantly, includes not only PSD data, but validation with other variables as much as feasible, including phytoplankton C, HPLC, and optical variables as you have helpfully suggested. In short, validation effort will continue and does not stop with this manuscript.

2. As Organelli et al., 2018 have shown, the same parameters retrieved here could be used to predict the beam attenuation measured by, for example, the LISST, C-star and AC-S instruments. We have many more data of all those from the whole ocean (e.g. Tara datasets on SeaBASS). Why not use them as part of your validation? It could help you to better constrain model parameters (particularly associated with NAP).

We thank you for this very helpful suggestion! In fact, as mentioned above, further validation is planned, and the full potential of all modeled variables by the novel algorithm has not been realized yet, it represents a large additional effort. For example, exploring bbp due to NAP. More importantly, we suggest as stated earlier that qualitative improvements in approach are needed in our opinion, and that importantly includes making more variables part of the retrieval, not just validation. For example, we've been strategizing the inclusion of phytoplankton and/or total particulate absorption in the inversion, rather than just bbp, i.e. a "blending" wit the approach of Roy et al. (2017).

3. Chlorophyll itself is a predictor of PFT, PCT and PSD. I would like to see, as I discussed with the lead author in the past, proof that the prediction of this model are significantly different than relationship with Chlorophyll itself or other current product from Rrs (e.g. bbp). Why go for a complicated model if a simple one is just as good (or just as bad, depending on your point of view)? I would love to see property-property plots involving C_phyto/POC, Chl, No and \xsi. In short, does the inclusion of all the model parameters into a novel model able to teach us about the ocean in ways [chl] does not (all the global maps presented seem highly correlated with [chl] and/or bbp distribution in the upper ocean)?

We recognize the fact that many variables in the ocean are correlated or strongly correlated with chlorophyll, especially the open ocean. This makes the issue of degrees of freedom and additional information content in more variables indeed an important one, especially in the context of many satellite variables being retrieved from a single multispectral spectrum. However, we also believe that it is not a good approach to push a more empirically oriented, reductionist agenda, where all things in the ocean are basically a function of Chl. While this is a very valid and practical approach, the overarching theme of this paper is to stay as mechanistic as possible (and push the envelope of what's possible to get from multispectral Rrs), even at the cost of a) degraded performance with respect to established, simpler, band-ratio algorithms or similar, b) the modeling and algorithm presented not being ideal for present-day multispectral data, i.e. its full potential is not realized with multispectral satellite data. We appreciate the fact that hyperspectral data is also only expected to have limited degrees of freedom, but even the ability to retrieve, say 2 or 3 more independent variables from hyperspectral Rrs will be a substantial improvement, i.e. perhaps we can model NAP and phyto PSD separately.

We also do not fully agree with everything is a function of chlorophyll approach for another reason, which is illustrated by the figures below (requested by the reviewer). While strong correlations with Chl do exist, there is substantial variability of the other parameters for a single Chl value. Brief discussion of these figures elaborating on this is given below. We will include some of these figures in the manuscript or its Supplementary material, and add a brief discussion of these figures to the main text.

[Figure]

The Chl to phyto C bivariate histogram for May 2015 (the example image used in our manuscript), shown above, indeed illustrates that a strong correlation with Chl exists, and this is expected. However, substantial variability of Phyto C for a single Chl value also exists, albeit admittedly a lot of the variability falls within estimated error bounds. Regardless, the Phyto C to Chl ratio is expected to vary in nature, as a results of physiological adaptation of phytoplankton, and is itself a central parameter of ocean ecological and biogeochemical modeling (e.g. Geider et al., 1998; Behrenfeld et al., 2006; Sathyendranath et al., 2020). Therefore, investigating the Chl to phyto C ratio in our model is a very important next step of verification and improvement and we thank you for this comment, indirectly emphasizing that.

[Figure]

[Figure]

The Chl to PSD parameters bivariate histograms are shown above. They illustrate that while a relationship obviously exists (with relatively high linear $R^2$ of about 0.8), substantial variability of the PSD parameters exists for a single value of Chl, especially so for the PSD slope. This illustrates that the PSD parameters and Chl are not "one and the same thing", and we do not advocate for juts retrieving the PSD via Chl. See also Brewin et al. (2012), their Fig. 6.

Below, we also show the relationship of the Stramski et al. (2008) POC retrievals vs. the PSD parameters, also illustrating the same conclusion – strong correlations with POC do exist, but substantial variability of the PSD parameters for a fixed POC value is present as well, especially for the PSD slope.

[Figure]

[Figure]

We of course do not claim that the x and y axes on these bivariate histograms are fully independent measurements. They are not because they both come from one set of Rrs spectra, and because they're not fully independent in oceanic ecosystems as well.

Finally, below we show the Stramski et al. (2008) POC vs. OC-CCI v. 5.0 Chl relationship. The relationship is stronger ($R^2 = 0.96$) that those shown above, showing the high predictability of POC from Chl. Yet, the community finds POC retrievals valuable as well, and both are considered standard OC products.

[Figure]

See also the similar bivariate histograms in Kostadinov et al. (2016). In conclusion, we agree that there is correlation with Chl, but we do not agree that there is no added value to our products. We do agree that there is a need for further investigation to avoid uniqueness of retrieval issues and

degrees of freedom/independence issues, more comprehensive and complete error propagation, and very clear communication to the community about the status of the algorithm. It is to be expected for many variables we retrieve from ocean color data to be correlated – after all they all come from one Rrs spectrum with 5 or 6 data points. We will add language in the manuscript to emphasize the points discussed above.

4. It is obvious that No and \xsi are correlated. Why not show their relationship? Is it consistent with in-situ data?

Thank you for suggesting this plot and analysis, it is informative, see below. Bivariate histograms of $\xi$ vs. $N_0$ are shown below (using the climatological OC-CCI v5.0-based sinusoidal projection image), with overlaid linear regressions for both the satellite (red) and the in-situ data (black crosses & regression line). The in-situ data is show after the duplicates averaging you suggested in your later comment, which is why the number of match-ups (N=167) is slightly lower than the one in the original manuscript.

[Figure]

When only a few outliers are removed, the in-situ data exhibit no significant linear relationship (slope 95% confidence interval crosses zero, $R^2 << 0.01$) between the two PSD parameters. See the figure below with outliers removed. We conclude that the subset of in-situ data matched with satellite data do not exhibit a significant relationship between x and N0, much unlike the satellite data, which do exhibit a strong negative correlation.

[Figure]

One significant observation is that satellite and in-situ PSD parameters do fall in similar ranges. While the 2.5 to 6 PSD slope range is prescribed in our model, this is particularly impressive for $N_0$, which is retrieved via the magnitude of modeled and observed bbp, and is not prescribed. This is an indication, that at least to first order or so, the algorithm is able to estimate the abundance of particles correctly. Given the large sources of uncertainty inherent in, for example, not knowing the index of refraction of particles accurately (and this is part of the Monte Carlo assessment of uncertainty), this is an encouraging result.

Contrary to the in-situ data, the satellite data exhibit strong negative $\xi$ to $N_0$ correlation. One possible interpretation of this is that physical reality in the global ocean indeed does not show this relationship in and the satellite observation is incorrect, an artifact of the modeling. However, we posit that this is not too likely, because a) satellite bbp exhibits steeper spectral slopes in oligotrophic waters (e.g. Loisel et al., 2006), and from first principles this is due to smaller particles dominance, and b) from ecological principles we know that oligotrophic waters are characterized by smaller organisms, e.g. dominance of cyanobacteria in the subtropical gyres. Therefore globally over the entire ocean and on average we would expect a clear negative correlation between x and $N_0$. This is indeed observed in the satellite data, as shown above. We note that the algorithm does not preclude retrievals that could show the opposite relationship – e.g. a place with very high abundance of very small particles vs. a place with very low abundance of relatively larger particles. It's just that the global ocean does not exhibit this relationship.

Next, we note the limited and likely biased spatio-temporal coverage of the in-situ data and posit that one possible reason for the lack of the correlation in in-situ data is indeed this limited coverage. Indeed, the lead author recollects a conference presentation from a while ago demonstrating the opposite x to $N_0$ correlation in in-situ data from the Gulf of Maine, similarly to the in-situ data presented here (with the outliers). The satellite data of course has uncertainties and assumptions, and we also note that the in-situ data comes from multiple instruments and is fitted over a relatively narrow size range. Therefore, discrepancies between in-situ PSD data at a single point and satellite data are expected.

5. The use of the same satellite matchup for multiple validation seems not acceptable to me as it is not an independent evaluation. You could/should average the in-situ data prior to matching up.

We have averaged the in-situ data ($\xi$, log10($N_0$), POC and pico-phyto C) for these duplicates and updated the validation plots to be used in the revised manuscript (see below). The overall scientific results and conclusions from these validation statistics have not changed. For the PSD N=177 before, whereas after duplicate removal N = 167; for POC: N = 4,238 vs N = 3,415, and for pico phyto C: N = 412 vs. N = 408 before and after, respectively. This illustrates that the most substantial reduction in number of match-ups was with POC, and for the other two variables it's minimal. We will add language to the methods to reflect this new duplicate removal.

[Figure]

[Figure]

[Figure]

Minor comments.

1. Figures numbers are not consistent with order of their citation.

Thank you for noting that, wherever possible (without disrupting logical flow for text and order of figures), this will be fixed in the revision.

2. Line 1445: remove 'using'.

Thank you for catching this (line 145), fixed.

References:

Behrenfeld, M. J., O'Malley, R. T., Siegel, D. A., McClain, C. R., Sarmiento, J. L., Feldman, G. C., Milligan, A. J., Falkowski, P. G., Letelier, R. M., & Boss, E. S. (2006). Climate-driven trends in contemporary ocean productivity. *Nature*, *444*(7120), 752–755. https://doi.org/10.1038/nature05317

Brewin, R. J. W., Dall'Olmo, G., Sathyendranath, S., & Hardman-Mountford, N. J. (2012). Particle backscattering as a function of chlorophyll and phytoplankton size structure in the open-ocean. *Opt. Express*, *20*(16), 17632–17652. https://doi.org/10.1364/OE.20.017632

Kostadinov, T. S., Siegel, D. A., & Maritorena, S. (2009). Retrieval of the particle size distribution from satellite ocean color observations. *Journal of Geophysical Research: Oceans*, *114*(9). https://doi.org/10.1029/2009JC005303

Geider, R. J., MacIntyre, H. L., & Kana, T. M. (1998). A dynamic regulatory model of phyto-planktonic acclimation to light, nutrients, and temperature. *Limnology and Oceanography*, *43*(4), 679–694. https://doi.org/10.4319/lo.1998.43.4.0679

Kostadinov, T. S., Milutinovic, S., Marinov, I., & Cabré, A. (2016). Carbon-based phytoplankton size classes retrieved via ocean color estimates of the particle size distribution. *Ocean Science*, *12*(2). https://doi.org/10.5194/os-12-561-2016

Loisel, H., Nicolas, J. M., Sciandra, A., Stramski, D., & Poteau, A. (2006). Spectral dependency of optical backscattering by marine particles from satellite remote sensing of the global ocean. *Journal of Geophysical Research: Oceans*, *111*(9), 1–14. https://doi.org/10.1029/2005JC003367

Roy, S., Sathyendranath, S., & Platt, T. (2017). Size-partitioned phytoplankton carbon and carbon-to-chlorophyll ratio from ocean colour by an absorption-based bio-optical algorithm. *Remote Sensing of Environment*, *194*, 177–189. https://doi.org/10.1016/j.rse.2017.02.015

Sathyendranath, S., Platt, T., Kovač, Ž., Dingle, J., Jackson, T., Brewin, R. J. W., Franks, P., Marañón, E., Kulk, G., & Bouman, H. A. (2020). Reconciling models of primary production and photoacclimation [Invited]. *Applied Optics*, *59*(10), C100. https://doi.org/10.1364/ao.386252

Stramski, D., Reynolds, R. A., Babin, M., Kaczmarek, S., Lewis, M. R., Röttgers, R., Sciandra, A., Stramska, M., Twardowski, M. S., Franz, B. A., & Claustre, H. (2008). Relationships between the surface concentration of particulate organic carbon and optical properties in the eastern South Pacific and eastern Atlantic Oceans. *Biogeosciences*, *5*(1), 171–201. https://doi.org/10.5194/bg-5-171-2008

Using the rubric of EGU:

We thank you for sharing the rubric responses with us. Our relevant replies to your rubric comments are provided above, or we explicitly respond below where needed.

Scientific significance -3

Scientific quality -3

Presentation quality – 2

1. Does the paper address relevant scientific questions within the scope of OS?

yes

2. Does the paper present novel concepts, ideas, tools, or data?

Yes.

3. Are substantial conclusions reached?

Not really, in my mind, beyond those of model evaluation.

We respectfully disagree that substantial conclusions are not reached. In fact, one important conclusion is that further significant leap in improvement is likely to require a substantial qualitative change in approach – e.g. one or more of – different PSD parameterization, inclusion of absorption (and/or other IOPs) in algorithm development and inversion, and regionalization of optical model inputs, e.g. indices of refraction (to address biases). Another important conclusion is that other existing algorithms/methods exhibit significant disagreements in terms of phyto C, making the need for $N_0$ tuning ambiguous.

4. Are the scientific methods and assumptions valid and clearly outlined?

Clearly outlined, but validity needs further work.

See our comments on validation above, where we discuss the $\xi$ vs. $N_0$ relationship, per your suggestion.

5. Are the results sufficient to support the interpretations and conclusions?

No.

See our responses to your previous comments.

6. Is the description of experiments and calculations sufficiently complete and

precise to allow their reproduction by fellow scientists (traceability of results)?

Yes.

7. Do the authors give proper credit to related work and clearly indicate their own

new/original contribution?

Yes.

8. Does the title clearly reflect the contents of the paper?

Yes.

9. Does the abstract provide a concise and complete summary?

Yes.

10. Is the overall presentation well structured and clear?

Yes.

11. Is the language fluent and precise?

Yes.

12. Are mathematical formulae, symbols, abbreviations, and units correctly defined

and used?

Yes.

13. Should any parts of the paper (text, formulae, figures, tables) be clarified,

reduced, combined, or eliminated?

See above comments.

See above our responses to your suggestions. Thank you for these constructive and helpful suggestions.

14. Are the number and quality of references appropriate?

Yes.

15. Is the amount and quality of supplementary material appropriate?

Yes.

---

## Referee Report (RR1)

**Review of revised manuscript egusphere-2022-430-ATC1**
25 March 2023

**General comment**

This study attempts to retrieve from space (ocean color satellite data) information on particle size distribution and carbon-based phytoplankton size classes in open ocean waters. This significant piece of work is actually the extension of previous studies (Kostadinov et al. 2007-2022) which includes validation results. The manuscript is well organized, written and illustrated.

Two reviewers pointed out validation results are not convincing, as several assumptions made in the methodology have limitations.

The authors have acknowledged these limits in their study but claim the validation results obtained are good enough considering their objective is to develop an operational algorithm for current multi-spectral satellite data. Authors have extended their validation datasets (addressing Rev#2 suggestion) and significantly modified the text to highlight the limits associated to their algorithm. Notably new references are presented and discussed, and a new section 'Further Discussion, Summary, and Conclusions" section summarizes the model limitations to future potential users.

I believe the revised manuscript accounts for most of the comments made by the reviewers and significantly improves the original submission. I therefore recommend the publication of this revised version (after correcting several typos in the text).

---

## Author Response (AR2)

Tihomir S. Kostadinov
Associate Professor of Geography
Dept. of Liberal Studies
CSU San Marcos, CA, USA

April 9, 2023

Dear Editor,

I am delighted to learn that our manuscript has been accepted. Thank you for your time and work during the peer review process. I reviewed the text for typos and will do so again carefully at the proofs stage. I thank the anonymous reviewer of the revised manuscript for their comments, to which, in my opinion, no further response is needed.

Best regards,

Tihomir Kostadinov, lead author.